# Regret Bounds for Non-decomposable Metrics with Missing Labels

**Nagarajan Natarajan**
Microsoft Research, INDIA
t-nanata@microsoft.com

**Prateek Jain**
Microsoft Research, INDIA
prajain@microsoft.com

## Abstract

We consider the problem of recommending relevant labels (items) for a given data point (user). In particular, we are interested in the practically important setting where the evaluation is with respect to non-decomposable (over labels) performance metrics like the $F_1$ measure, *and* training data has missing labels. To this end, we propose a generic framework that given a performance metric $\Psi$, can devise a regularized objective function and a threshold such that all the values in the predicted score vector above and only above the threshold are selected to be positive. We show that the regret or generalization error in the given metric $\Psi$ is bounded ultimately by estimation error of certain underlying parameters. In particular, we derive regret bounds under three popular settings: a) collaborative filtering, b) multilabel classification, and c) PU (positive-unlabeled) learning. For each of the above problems, we can obtain precise non-asymptotic regret bound which is small even when a large fraction of labels is missing. Our empirical results on synthetic and benchmark datasets demonstrate that by explicitly modeling for missing labels and optimizing the desired performance metric, our algorithm indeed achieves significantly better performance (like $F_1$ score) when compared to methods that do not model missing label information carefully.

## 1 Introduction

Predicting relevant labels/items for a given data point is by now a standard task with applications in several domains like recommendation systems [Koren et al., 2009], document tagging, image tagging [Prabhu and Varma, 2014], etc. In many problem settings, e.g. collaborative filtering, features for the data points might not be available and one needs to predict labels only on the basis of past labels (e.g., existing likes/dislikes for various labels/items). In the presence of features, the prediction problem is the standard multi-label classification problem.

Design and analysis of algorithms for such tasks should counter two fundamental challenges: a) in practical scenarios, desired performance metrics for predictions are typically complex *non-decomposable* functions such as $F_1$ score or precision@$k$; standard metrics like Hamming loss or RMSE over the labels may not be useful, and b) any realistic system in this domain should be able to handle missing labels. Furthermore, often the locations of missing labels may not be available; an important special case is the positive-unlabeled learning setting [Hsieh et al., 2015]. Dealing with missing labels may necessitate imposition of certain regularization on the parameters such as low-rank regularization so as to exploit the correlations between labels.

Most of the existing solutions address only one of the two aspects. For example, Koyejo et al. [2015] establish that for a large class of performance metrics, the optimal solution is to compute a score vector over all the labels and selecting all the labels with score greater than a constant. Their algorithm treats each label as independent to estimate class-conditional probability separately for each label. Clearly, such methods ignore available information about other labels, and hence cannot handle

missing information effectively. Also, such methods do not even apply to the collaborative filtering setting. On the other hand, most of the existing collaborative filtering/matrix completion methods focus only on decomposable losses like RMSE, sum of logistic loss [Lafond, 2015, Yu et al., 2014], which are not effective in real-world systems with large number of labels [Prabhu and Varma, 2014].

In this work, we devise a simple and generic framework that addresses both the aforementioned issues; the framework leads to simple and efficient algorithms in several different settings and for a wide variety of performance metrics used in practice including the multi-label $F$-measure. Our framework is motivated by a simple observation that has been used in other contexts as well [Kotłowski and Dembczyński, 2015, Koyejo et al., 2015]: for a large class of metrics $\Psi$, simply thresholding the class probability vector leads to bayes-optimal estimators. Hence, the goal would be to estimate per-label class probabilities accurately. To this end, we show that using a strongly proper loss along with appropriate thresholding leads to bounded regret wrt. $\Psi$ (Theorem 1), and that the threshold can be learned using cross-validation over a small fraction of the training data.

In more detail, strong convexity of the loss function ensures that by minimizing a nuclear-norm regularized ERM (with risk measured by the selected loss function) wrt. a parameter matrix $W \in \mathbb{R}^{d \times L}$, we can bound the regret in $\Psi$ by regret in estimation of the optimal $W$ (Theorem 1); here, $d$ is the dimensionality of the data and is equal to number of users in case of recommender system. Hence, this result allows us to focus on estimation of $W^*$ in various settings such as: a) one-bit matrix completion (Theorem 2), popularly used in recommender systems with only like/dislike information, b) one-bit matrix completion with PU learning (Theorem 4) applicable to recommender systems where only "likes" or positive feedback is observed, and c) general multi-label learning with missing labels (Theorem 3).

For one-bit matrix completion (and the related PU setting), we obtain our final regret bound by adapting existing results from Lafond [2015] and Hsieh et al. [2015], respectively. For general multilabel setting, a direct application of existing results, such as [Lafond, 2015] leads to *weak* bounds. A main technical contribution of our work is to analyze the parameter estimation problem in this setting and provide tight regret bounds. In fact, our result strictly generalizes the result by Lafond [2015], which is for matrix completion with exponential family noise, to the general *inductive matrix completion* setting [Jain and Dhillon, 2013] with exponential family noise. Hence, it should have applications beyond our framework as well. Finally, we illustrate our framework and algorithms on synthetic as well as real-world datasets. Our method exhibits significant improvement over a natural extension of the method by Koyejo et al. [2015] that optimizes $\Psi$ directly but ignores label correlations, hence does not handle missing labels in a principled manner. For example, our method achieves 12% higher $F_1$-measure on a benchmark dataset than that by Koyejo et al. [2015].

**Related Work.** We now highlight some related theoretical work in recommender systems and multi-label learning. Gao and Zhou [2013] study consistency and surrogate losses for two specific losses namely Hamming and expected (partial) ranking losses, and leave the other losses to future work. Dembczynski et al. [2012] consider expected pairwise ranking loss in multilabel learning, show that the problem decomposes into independent binary problems, and provide regret bound for the same. Yun et al. [2014] consider the learning-to-rank problem, where the goal is to rank the relevant labels for a given instance. They show that popular ranking losses like NDCG can be written as a generalization of certain robust binary loss functions, although they do not provide any explicit regret bounds. Existing theoretical guarantees for 1-bit matrix completion methods used in recommender systems focus solely on RMSE or 0-1 loss [Lafond, 2015, Hsieh et al., 2015].

## 2   Problem Setup and Background

Let $\mathbf{x}_i \in \mathcal{X} \subseteq \mathbb{R}^d$ denote instances and $\mathbf{y}_i \in \{0,1\}^L$ denote label vectors. Let $\mathsf{Y} \in \{0,1\}^{n \times L}$ denote the label matrix, with $\mathbf{y}_i$'s as rows. In typical multi-label learning and recommender system settings a) the labeling process has some inherent uncertainty, which is usually captured by assuming a conditional distribution $\mathbb{P}(\mathbf{y}_i|\mathbf{x}_i)$, b) furthermore, we do not get to observe all the entries of $\mathbf{y}_i$, but only a small subset, say $\Omega_i$. Formally, let $\Omega \subset [n] \times [L]$ denote a subset of indices sampled i.i.d. from a fixed distribution $\pi$ over $[n] \times [L]$. We consider the following sampling model for observing labels:

$$\mathsf{Y}_{ij} = \begin{cases} 1 & \text{with probability } g_j(\mathbf{x}_i; W^*) \\ 0 & \text{with probability } 1 - g_j(\mathbf{x}_i; W^*) \end{cases} \text{ for } (i,j) \in \Omega. \tag{1}$$

where $W^*$ parameterizes the underlying conditional distribution $\mathbb{P}(\mathbf{y}_i|\mathbf{x}_i)$. Following the low-rank *inductive* matrix completion model [Yu et al., 2014, Zhong et al., 2015], we let $W^* \in \mathbb{R}^{d \times L}$ be the parameter matrix and $g_j(\mathbf{x}_i; W^*) = g(\langle \mathbf{x}_i, \mathbf{w}_j^* \rangle)$ where $\mathbf{w}_j^*$ is the $j$th column of $W^*$ corresponding to the $j$th label, for some differentiable function $g : \mathbb{R} \to [0, 1]$. A popular choice of $g$ is given by $g(\langle \mathbf{x}_i, \mathbf{w}_j \rangle) = \frac{\exp(\langle \mathbf{x}_i, \mathbf{w}_j \rangle)}{1 + \exp(\langle \mathbf{x}_i, \mathbf{w}_j \rangle)}$, which corresponds to the logistic regression model. When we do not observe feature vectors $\mathbf{x}$, as in the classical recommender system or matrix completion setting, the above model (1) reduces to the widely studied 1-bit matrix completion model [Cai and Zhou, 2013, Davenport et al., 2014]:

$$\mathsf{Y}_{ij} = \begin{cases} 1 & \text{with probability } g(W_{ij}^*) \\ 0 & \text{with probability } 1 - g(W_{ij}^*) \end{cases} \text{ for } (i, j) \in \Omega, \tag{2}$$

where $W^* \in \mathbb{R}^{n \times L}$ is the parameter matrix that captures user-item preferences.

The goal is to learn a multi-label classifier $f : \mathcal{X}^n \to \{0, 1\}^{n \times L}$ jointly over $n$ instances, such that a performance metric of interest $\Psi$ is maximized. The training data consists of input features $\mathsf{X} \in \mathbb{R}^{n \times d}$ where each row corresponds to an instance, drawn iid from some distribution $\mathbb{P}_{\mathcal{X}}$ over $\mathcal{X}$, and *partially observed* label matrix $\mathsf{Y}$ using the sampling model (1) or (2). In this work, we consider a large family of non-decomposable metrics [Koyejo et al., 2015] that constitutes linear-fractional functions of (multi-label analogues of) true positives, false positives, false negatives and true negatives defined below. Let $\widehat{\mathsf{Y}} \in \{0, 1\}^{n \times L}$ denote the predicted labels, i.e. $\widehat{\mathsf{Y}} = f(\mathsf{X})$ for some $f$. Define the primitives:

$$\widehat{\mathrm{TP}}_{ij}(\widehat{\mathsf{Y}}, \mathsf{Y}) = [[\hat{\mathsf{Y}}_{ij} = 1, \mathsf{Y}_{ij} = 1]], \quad \widehat{\mathrm{FP}}_{ij}(\widehat{\mathsf{Y}}, \mathsf{Y}) = [[\hat{\mathsf{Y}}_{ij} = 1, \mathsf{Y}_{ij} = 0]],$$
$$\widehat{\mathrm{TN}}_{ij}(\widehat{\mathsf{Y}}, \mathsf{Y}) = [[\hat{\mathsf{Y}}_{ij} = 0, \mathsf{Y}_{ij} = 0]], \quad \widehat{\mathrm{FN}}_{ij}(\widehat{\mathsf{Y}}, \mathsf{Y}) = [[\hat{\mathsf{Y}}_{ij} = 0, \mathsf{Y}_{ij} = 1]].$$

For convenience, we drop the arguments and just write $\widehat{\mathrm{TP}}_{ij}$ to denote $\widehat{\mathrm{TP}}_{ij}(\widehat{\mathsf{Y}}, \mathsf{Y})$ and so on.

1. **Micro-averaged metrics.** Define: $\widehat{\mathrm{TP}}(\widehat{\mathsf{Y}}, \mathsf{Y}) = \frac{1}{|\Omega|} \sum_{(i,j) \in \Omega} \widehat{\mathrm{TP}}_{ij}$ and $\widehat{\mathrm{FP}}(\widehat{\mathsf{Y}}, \mathsf{Y}), \widehat{\mathrm{TN}}(\widehat{\mathsf{Y}}, \mathsf{Y}), \widehat{\mathrm{FN}}(\widehat{\mathsf{Y}}, \mathsf{Y})$ similarly. Let $\mathrm{TP} = \mathbb{E}[\widehat{\mathrm{TP}}], \mathrm{FP} = \mathbb{E}[\widehat{\mathrm{FP}}]$ (and so on), where the expectation is defined wrt to the sampling distribution $\pi$ over indices $[n] \times [L]$ as well as the joint distribution $\mathbb{P}$. Micro-averaged performance metric $\Psi : \{0, 1\}^{n \times L} \times \{0, 1\}^{n \times L} \to \mathbb{R}_+$ is given by:

$$\Psi(\widehat{\mathsf{Y}}, \mathsf{Y}) = \frac{a_0 + a_{11}\mathrm{TP} + a_{01}\mathrm{FP} + a_{10}\mathrm{FN} + a_{00}\mathrm{TN}}{b_0 + b_{11}\mathrm{TP} + b_{01}\mathrm{FP} + b_{10}\mathrm{FN} + b_{00}\mathrm{TN}}. \tag{3}$$

for bounded constants $a$'s and $b$'s. Assume that $\Psi$ is bounded, i.e. $\exists \gamma' > 0$ such that $b_0 + b_{11}\mathrm{TP} + b_{01}\mathrm{FP} + b_{10}\mathrm{FN} + b_{00}\mathrm{TN} > \gamma'$ for all $\widehat{\mathsf{Y}}, \mathsf{Y}$.

2. **Instance-averaged metrics.** Define $\widehat{\mathrm{TP}}_i(\widehat{\mathsf{Y}}, \mathsf{Y}) = \frac{1}{|\Omega_i|} \sum_{j \in \Omega_i} \widehat{\mathrm{TP}}_{ij}$. Let $\mathrm{TP}_i = \mathbb{E}[\widehat{\mathrm{TP}}_i]$. Instance-averaged performance metric $\Psi$ is given by:

$$\Psi(\widehat{\mathsf{Y}}, \mathsf{Y}) = \frac{1}{n} \sum_{i=1}^{n} \frac{a_0 + a_{11}\mathrm{TP}_i + a_{01}\mathrm{FP}_i + a_{10}\mathrm{FN}_i + a_{00}\mathrm{TN}_i}{b_0 + b_{11}\mathrm{TP}_i + b_{01}\mathrm{FP}_i + b_{10}\mathrm{FN}_i + b_{00}\mathrm{TN}_i}. \tag{4}$$

for bounded constants $a$'s and $b$'s. Assume that $\Psi$ is bounded, i.e. $\exists \gamma' > 0$ such that $b_0 + b_{11}\mathrm{TP}_i + b_{01}\mathrm{FP}_i + b_{10}\mathrm{FN}_i + b_{00}\mathrm{TN}_i > \gamma'$ for all $\widehat{\mathsf{Y}}, \mathsf{Y}, i$.

3. **Macro-averaged metrics.** Let $\Omega^{(j)} = \{i : (i, j) \in \Omega\}$. Define: $\widehat{\mathrm{TP}}_j(\widehat{\mathsf{Y}}, \mathsf{Y}) = \frac{1}{|\Omega^{(j)}|} \sum_{i \in \Omega^{(j)}} \widehat{\mathrm{TP}}_{ij}$. Let $\mathrm{TP}_j = \mathbb{E}[\widehat{\mathrm{TP}}_j]$. Macro-averaged performance metric $\Psi$ is given by:

$$\Psi(\widehat{\mathsf{Y}}, \mathsf{Y}) = \frac{1}{L} \sum_{j=1}^{L} \frac{a_0 + a_{11}\mathrm{TP}_j + a_{01}\mathrm{FP}_j + a_{10}\mathrm{FN}_j + a_{00}\mathrm{TN}_j}{b_0 + b_{11}\mathrm{TP}_j + b_{01}\mathrm{FP}_j + b_{10}\mathrm{FN}_j + b_{00}\mathrm{TN}_j}. \tag{5}$$

for bounded constants $a$'s and $b$'s. Assume that $\Psi$ is bounded, i.e. $\exists \gamma' > 0$ such that $b_0 + b_{11}\mathrm{TP}_j + b_{01}\mathrm{FP}_j + b_{10}\mathrm{FN}_j + b_{00}\mathrm{TN}_j > \gamma'$ for all $\widehat{\mathsf{Y}}, \mathsf{Y}, j$.

**Example metrics**:

1. Instance-averaged $F_1$ metric defined as: $\Psi_{F_1}(\widehat{\mathsf{Y}}, \mathsf{Y}) = \frac{1}{n} \sum_{i=1}^{n} \frac{2\mathrm{TP}_i}{2\mathrm{TP}_i + \mathrm{FP}_i + \mathrm{FN}_i}$.

2.  Accuracy (equivalent to the Hamming loss): $\Psi_{\text{Ham}}(\widehat{\mathsf{Y}}, \mathsf{Y}) = 1 - \frac{1}{n}\sum_{i=1}^{n}\text{FP}_i + \text{FN}_i$.

**Remark 1.** *The aforementioned definitions of performance metrics naturally apply to the recommender system setting, where data is observed via the 1-bit matrix completion sampling model* (2). *Note that in this case, the expectations are defined wrt the sampling distribution $\pi$ and the inherent noise in 1-bit sampling $\mathbb{P}(\mathsf{Y}_{ij}|W_{ij}^*)$.*

Let $\Psi^*$ denote the Bayes optimal performance, i.e. $\Psi^* = \max_f \Psi(f(\mathsf{X}), \mathsf{Y})$. Our objective can be now stated learning $\hat{f}$ such that the $\Psi$-regret, i.e. $\Psi^* - \Psi(\hat{f}(\mathsf{X}), \mathsf{Y})$, is provably bounded. Koyejo et al. [2015] showed that the Bayes optimal $\Psi^*$ thresholds the conditional probability of each label $j$, i.e. $\mathbb{P}(y_j|\mathbf{x})$ at a certain value $\delta^* \in (0, 1)$, and that the value $\delta^*$ is shared across all the labels.[1]:

# 3  Algorithm

Our approach is based on estimating real-valued predictions and then thresholding the predictions optimally in order to maximize a given metric $\Psi$. Koyejo et al. [2015] proposed a simple consistent plug-in estimator algorithm, which first computes conditional marginals $\mathbb{P}(y_j|\mathbf{x})$ independently for each label $j$, and then estimates a threshold jointly to optimize $\Psi$. While the approach is provably consistent asymptotically, it is not clear if it admits a useful regret bound; in particular, we would like to characterize the behavior in the finite samples regime. In case of the sampling model (1), the approach translates to learning columns of the parameter matrix $\mathsf{W}^*$ independently. In many cases, $\mathsf{W}$ exhibits some structure, such as low-rankness, reflecting correlation between labels [Yu et al., 2014, Zhong et al., 2015, Davenport et al., 2014]. Statistically, capturing correlations via a low-rank structure could help improve the sample complexity for recovery, and computationally, it would help reduce space and time complexity of the learning procedure.

Our proposed algorithm is presented in Algorithm 1. In Step 1, we solve a trace-regularized minimization problem to estimate the parameter matrix $\mathsf{W}^*$, where the function $\ell$ can be any bounded loss such as the squared, the logistic or the squared Hinge loss. In particular, using the logistic loss corresponds to the maximum likelihood estimation of the sampling model (1). Yu et al. [2014] also solve essentially the same objective as (6), except for the additional bound constraint on entries of $\mathsf{XW}$. The optimization problem (6) can be solved using a proximal gradient descent algorithm, with a fast proximal operator computation by storing the current solution in a low-rank form. We could also use fast non-convex procedure, by writing $\mathsf{W} = \mathsf{W}_1 \mathsf{W}_2^T$, where $\mathsf{W}_1$ and $\mathsf{W}_2$ are low-rank matrices with $k \ll \min(d, L)$ columns each, and applying alternating minimization.

The real-valued estimator is given by $\mathsf{Z} = \mathsf{X}\hat{\mathsf{W}}$ in Step 2. To obtain binary-valued predictions, we solve a 1-dimensional optimization problem to compute the optimal threshold, on the training data. Note that this step can be done in $|\Omega|$ time.

**Remark 2.** *In the 1-bit matrix completion setting, we obtain a thresholded max-likelihood estimator of $W^* \in \mathbb{R}^{n \times L}$ using identical procedure; where we interpret $\mathsf{X}$ in Algorithm 1 as the identity matrix of size $n$.*

# 4  Analysis: Regret Bounds

Here, we first show that $\Psi$-regret can be bounded with the regret of a certain loss $\ell$. Then, under various sampling models pertaining to different settings, namely, 1-bit matrix completion, multi-label learning, and PU (positive-unlabeled) learning, we show that the $\ell$-regret can be bounded via recovering the underlying parameter matrix $\mathsf{W}^*$ governing $\mathbb{P}(\mathbf{y}_i|\mathbf{x}_i)$.

## 4.1  Low $\ell$-regret implies low $\Psi$-regret

Our first main result connects $\Psi$-regret to regret with respect to a strongly proper loss function $\ell$ [Agarwal, 2014]. Canonical examples of strongly proper losses include the logistic loss $\ell(t, y) = \log(1 + \exp(-(2y - 1)t))$, the exponential loss $\ell(t, y) = \exp(-(2y - 1)t)$ and the squared loss

**Algorithm 1** Thresholded Max-Likelihood Estimator

---

**Input**: Training data $\mathsf{X} \in \mathbb{R}^{n \times d}$, labels $\mathsf{Y}_\Omega \in \{0,1\}^{n \times L}$ observed on indices $\Omega$, metric $\Psi$, parameters $\gamma, \lambda$.

1. Obtain $\hat{\mathsf{W}}$ by solving the trace-constrained matrix completion:

$$\hat{\mathsf{W}} = \arg \min_{\mathsf{W}: \|\mathsf{X}\mathsf{W}\|_\infty \leq \gamma} \frac{1}{|\Omega|} \sum_{(i,j) \in \Omega} \ell(\langle \mathbf{x}_i, \mathbf{w}_j \rangle, \mathsf{Y}_{ij}) + \lambda \|\mathsf{W}\|_*, \qquad (6)$$

2. Let $\mathsf{Z} = \mathsf{X}\hat{\mathsf{W}}$. Define the thresholding operator $\widehat{\mathsf{Y}} = \mathrm{Thr}_\theta(\mathsf{Z})$, such that $\hat{\mathsf{Y}}_{ij} = [\![\mathsf{Z}_{ij} \geq \theta]\!]$.
3. Let $\widehat{\Psi}$ denote the empirical metric computed on a finite sample; i.e., (3), (4), (5) defined wrt. $\widehat{\mathrm{TP}}_{ij}$, $\widehat{\mathrm{FP}}_{ij}$, $\widehat{\mathrm{TN}}_{ij}$ and $\widehat{\mathrm{FN}}_{ij}$.
4. Return $\widehat{\mathsf{Y}} = \mathrm{Thr}_{\hat\theta}(\mathsf{Z})$, where

$$\hat{\theta} = \arg \max_\theta \widehat{\Psi}(\mathrm{Thr}_\theta(\mathsf{Z}_\Omega), \mathsf{Y}_\Omega),$$

---

$\ell(t, y) = (1 - (2y - 1)t)^2$, for $y \in \{0,1\}$ and $t \in \mathbb{R}$. Define the $\ell$-regret of $\mathsf{Z} \in \mathbb{R}^{n \times L}$ as:

$$\mathrm{Reg}_\ell(\mathsf{Z}) = \mathbb{E}[\ell(\mathsf{Z}_{ij}, \mathsf{Y}_{ij})] - \min_{\mathsf{Z}' \in \mathbb{R}^{n \times L}} \mathbb{E}[\ell(\mathsf{Z}'_{ij}, \mathsf{Y}_{ij})],$$

where the expectation is wrt. draws from $\pi$ and the joint distribution $\mathbb{P}$ over instances and labels.

**Theorem 1** (Main Result 1). *Let $\Psi$ be a performance metric as defined in* (3)*,* (4) *or* (5)*. Let $\ell$ be a $\lambda'$-strongly proper loss function. Assume the input $\mathsf{X} \in \mathbb{R}^{n \times L}$ consists of iid instances sampled from marginal $\mathbb{P}_\mathcal{X}$, label matrix $\mathsf{Y}_\Omega \in \{0,1\}^{n \times L}$, where $\mathsf{Y}_{ij}$ is sampled iid from $\mathbb{P}(\mathsf{Y}_{ij}|\mathbf{x}_i)$, observed on a subset of indices $\Omega$ sampled iid from a fixed distribution $\pi$. Then, the output $\widehat{\mathsf{Y}}$ obtained by thresholding the estimate $\mathsf{Z}$ in Step 4 of Algorithm 1 satisfies the regret bound:*

$$\Psi^* - \Psi(\widehat{\mathsf{Y}}, \mathsf{Y}) \leq C \sqrt{\frac{2}{\lambda'}} \sqrt{\mathrm{Reg}_\ell(\mathsf{Z})} + O\left(\frac{1}{\sqrt{|\Omega|}}\right), \qquad (7)$$

*for some positive constant $C$.*

We emphasize that the above result holds for arbitrary metric $\Psi$ from the family (3), (4) or (5). Consider the RHS of (7): $1/\sqrt{|\Omega|}$ is the lower-order term, and independent of dimensionality; the first term makes the framework fairly powerful, as it can use any strongly proper loss. In the next subsection, we will provide precise instantiations of this term under various learning settings.

**Proof Outline for Theorem 1.** Proof technique is based on [Kotłowski and Dembczyński, 2015], where they derive similar bound in the binary classification setting. We first relate the $\Psi$-regret to weighted 0-1 loss regret (Lemma 2). Then, we show there exists a threshold $\theta^*$ such that the weighted loss regret of $\mathrm{Thr}_{\theta^*}(\mathsf{Z}) \in \{0,1\}^{n \times L}$ is bounded by its $\ell$-regret corresponding to any strongly proper loss $\ell$ (Lemma 3). Finally, we argue that it suffices to estimate $\hat{\theta}$ from the training data (Lemma 4). Detailed proof and associated Lemmas are available in Appendix A.1. $\qquad \square$

## 4.2 Bounding $\ell$-regret

Below, we provide the desired $\ell$-regret bound under three different settings.

### 4.2.1 Collaborative Filtering

Consider the 1-bit matrix completion sampling model in (2). Then (6) reduces to the optimization problem considered by Lafond [2015]. We have the following regret bound for the estimator $\mathsf{Z} = \hat{\mathsf{W}}$ obtained in Step 2 of Algorithm 1 (Note that $\mathsf{X}$ is just treated as identity in this setting).

**Theorem 2.** *Assume $\pi$ is uniform, and consider the 1-bit matrix completion sampling model* (2)*. Let $\ell$ denote a 1-Lipschitz, strongly proper loss (appearing in* (7)*), and $\mathsf{Z}$ denote the output of Step 2 of*

*Algorithm 1 with the choice of* $\lambda = 2c_\gamma \sqrt{\frac{2\log(n+L)}{\min(n,L)|\Omega|}}$. *With probability at least* $1 - \delta$, *the following holds:*

$$\mathrm{Reg}_\ell(\mathsf{Z}) \leq \sqrt{\tilde{C} \max\left(\frac{\max(n,L)\, \mathsf{rank}(W^*)\log(3/\delta)}{|\Omega|}\left(\sigma_\gamma^2 + 1\right), \gamma^2 \sqrt{\frac{\log(3/\delta)}{|\Omega|}}\right)},$$

*where* $\tilde{C}, c_\gamma, \sigma_\gamma$ *are numerical constants, and* $\gamma = \max_{ij}|W_{ij}^*|$.

Note that when $|\Omega| > \max(n,L)$, the RHS of the above bound converges; in particular, the second term within $\max$ is the lower-order term: $\gamma \approx O\big(\sqrt{1/nL}\big)$. Theorem 2 can be extended to general distributions $\pi$ beyond uniform, satisfying mild assumptions. See Appendix A.2.

### 4.2.2 Multi-label Learning

Consider the sampling model (1) with features. We have the following regret bound for the estimator $\mathsf{Z} = \mathsf{X}\hat{\mathsf{W}}$ obtained in Step 2 of Algorithm 1, under the following assumptions.

**Assumption 1.** *The marginal distribution over the features* $\mathbb{P}_\mathcal{X}$ *is sub-Gaussian with sub-Gaussian norm* $K$ *and covariance* $\Sigma \in \mathbb{R}^{d\times d}$.

**Assumption 2.** *Let* $\pi_{k,l}$ *denote the probability of sampling the entry* $(k,l) \in [n] \times [L]$; $\exists\, \mu \geq 1$ *s.t. for all* $n, L$, $\min_{k\in[n],l\in[L]} \pi_{k,l} \geq \frac{1}{\mu n L}$, *and* $\nu \geq 1$ *s.t.* $\max_{i',j'}\left(\sum_j \pi_{i'j}, \sum_i \pi_{ij'}\right) \leq \frac{\nu}{\min(n,L)}$.

**Theorem 3** (Main Result 2). *Assume 1, 2 and consider the sampling model* (1). *Also assume* $L \geq d$. *Let* $\hat{W}$ *be the solution to the trace-norm regularized optimization problem* (6) *using logistic loss for* $\ell$, *number of training data points* $n \geq C'd$, *number of observations* $|\Omega| \geq L + d$, *and setting the regularization parameter* $\lambda = \frac{2c}{\sqrt{|\Omega|}}$. *Then, with probability at least* $1 - 3(n+L)^{-1} - 2(d+L)^{-1}$, *the following holds:*

$$\frac{\|\hat{W} - W^*\|_F^2}{dL} \leq \frac{C_2 \mu^2}{d} \max\left(\frac{L\, \mathsf{rank}(W^*)\log(n+L)}{|\Omega|}\left(\sigma_\gamma^2 + 1\right), \frac{\gamma^2}{\mu}\sqrt{\frac{\log(n+L)}{|\Omega|}}\right),$$

*where* $c, C', C_2$ *are numerical constants,* $\sigma_\gamma \leq (1 + e^\gamma)^2 e^\gamma$, $\gamma = \max_{ij}|(XW^*)_{ij}|$ *and* $\mu$ *is defined as in Assumption 2.*

A few remarks of our result in the multi-label setting are in order:

**Remark 3** (Generalization). *The result in Theorem 3, and Theorem 8 in Appendix B for general exponential distributions, is a key technical contribution of this work. In particular, our analysis applies to* $Y$ *arising from general exponential distributions, including Gaussian when* $Y$ *is real-valued and Poisson when* $Y$ *models counts. See Appendix B for more details.*

**Remark 4** (Comparing [Lafond, 2015]). *If we directly apply the method and the analysis of [Lafond, 2015], the resulting bounds are very weak; in fact, when* $n \geq L$ *and* $|\Omega| = O(n)$, *which is quite common in the multi-label scenario, the ensuing bound suggests that the estimator is not even consistent, even when* $\pi$ *is uniform. See Appendix A.3 for details.*

**Remark 5** (Comparing [Koyejo et al., 2015]). *The plugin-in estimator algorithm of [Koyejo et al., 2015] estimates* $\mathbf{w}_j^*$ *for each label* $j$ *independently, and learns a common threshold as in Step 4 of Algorithm 1. Let* $\hat{\mathbf{w}}_j$ *denote the estimator for label* $j$. *Then, using standard analysis we have,* $\|\hat{\mathbf{w}}_j - \mathbf{w}_j^*\|_2 \leq \sigma\sqrt{\frac{d}{|\Omega^j|}}$, *where* $|\Omega^j|$ *is the number of observations per label which is* $O(\frac{|\Omega|}{L})$. *Thus we have the bound:* $\frac{\|W^* - \hat{W}\|_F^2}{L} \leq \sigma O(\frac{Ld}{|\Omega|})$. *This is how our bounds behave, when* $W^*$ *is indeed full rank, up to constants. When* $\mathsf{rank}(W^*) \ll \min(d, L)$, *we achieve much faster convergence.*

We now give the desired $\ell$-regret bound as a corollary.

**Corollary 1.** *Assume the conditions and the choices in Theorem 3 hold. Let* $\ell$ *denote a 1-Lipschitz, strongly proper loss (appearing in* (7)*), and* $\mathsf{Z} = \mathsf{X}\hat{\mathsf{W}}$ *denote the output of Step 2 of Algorithm 1. With probability at least* $1 - \delta - (d+L)^{-1}$, *the following holds:*

$$\mathrm{Reg}_\ell(\mathsf{Z}) \leq \sqrt{C_2 \mu^2 \max\left(\frac{L\, \mathsf{rank}(W^*)\log(3/\delta)}{|\Omega|}\left(\sigma_\gamma^2 + 1\right), \frac{\gamma^2}{\mu}\sqrt{\frac{\log(3/\delta)}{|\Omega|}}\right)},$$

*where* $c, C', C_2, \sigma_\gamma, \gamma, \mu$ *are defined as in Theorem 3.*

**Proof Outline for Theorem 3.** We analyze the following general exponential noise model for $\mathsf{Y}$:

$$\mathsf{Y}_{ij}|\mathbf{x}_i, \mathbf{w}_j \sim \exp_{h,G}(\mathbf{x}_i, \mathbf{w}_j) := h(\mathsf{Y}_{ij}) \exp\left(\langle \mathbf{x}_i, \mathbf{w}_j \rangle \mathsf{Y}_{ij} - G(\langle \mathbf{x}_i, \mathbf{w}_j \rangle)\right), \qquad (8)$$

where $h$ and $G$ are the base measure and log-partition functions associated with this canonical representation. Our proof sketch is based on Lafond [2015], but requires bounding certain quantities carefully. In particular, we prove a tight bound for $\|\mathsf{X}^T \nabla \Phi_{\mathsf{Y}}(\mathsf{X}, \mathsf{W}^*)\|_2$ in terms of the regularization parameter $\lambda$, where $\Phi_{\mathsf{Y}}(\mathsf{X}, \mathsf{W}^*)$ is the MLE wrt. general exponential distribution (reduces to (6), without regularization, when $\mathsf{Y}_{ij}$'s are from (1)), as stated below.

**Lemma 1.** *Consider the sampling model* (8). *Assume (i)* $d \leq L$, *(ii)* $|\Omega| \geq (L + d)$, *(iii)* $\mathsf{Y}_{ij}$'s *are sampled independently given* $\mathbf{x}_i$, *and (iv)* $|\mathsf{Y}_{ij} - G'(\langle \mathbf{x}_i, \mathbf{w}_j^* \rangle)| \leq \alpha$, *for all* $i, j \in [n] \times [L]$, *for any* $n, L$. *Let* $\mathsf{X} \in \mathbb{R}^{n \times d}$ *whose rows* $(\mathbf{x}_i$'s*) are iid samples from* $\mathbb{P}_{\mathcal{X}}$ *satisfying Assumption 1. Then, there exists numerical constant* $c$ *such that, with probability at least* $1 - (d + L)^{-1}$:

$$\left\| \mathsf{X}^T \nabla \Phi_Y(\mathsf{X}, \mathsf{W}^*) \right\|_2 \leq c \cdot \frac{\alpha}{\sqrt{|\Omega|}}.$$

### 4.2.3 PU Learning

In many collaborative filtering and multi-label learning tasks, only the positive entries $(y_{ij} = 1)$ are observed. In this setting, we can use the approach of [Hsieh et al., 2015], where they consider a two-stage sampling model: sample $y_{ij}$ using (2) for all $i, j \in [n] \times [L]$ (or using (1) when features are available), and then flip a fraction $\rho$ of the sampled 1's to 0's, resulting in $\tilde{\mathsf{Y}}$. We would then use the unbiased estimator $\tilde{\ell}$ of loss $\ell$ in (6); $\tilde{\ell}$ satisfies $\mathbb{E}[\tilde{\ell}(\mathsf{Z}_{ij}, \tilde{\mathsf{Y}}_{ij})] = \ell(\mathsf{Z}_{ij}, \mathsf{Y}_{ij})$, where the expectation is wrt the flipping process, parameterized by $\rho$. For the estimator $\mathsf{Z} = \hat{\mathsf{W}}$ obtained thus, we have the following regret bound.

**Theorem 4.** *Let* $\ell$ *denote a 1-Lipschitz, strongly proper loss (appearing in* (7)*). Assume* $\|W^*\|_* \leq t$. *Let* $\mathsf{Z} = \mathsf{X}\hat{\mathsf{W}}$, *where* $\hat{\mathsf{W}}$ *is obtained by solving the unbiased estimator objective of Hsieh et al.* [2015]. *There exists a numerical constant* $C_3$ *such that, with probability at least* $1 - \delta$:

$$\text{Reg}_\ell(\mathsf{Z}) \leq \sqrt{6 \frac{\sqrt{\log(2/\delta)}}{\sqrt{nL}(1 - \rho)} + 2C_3 \cdot t \frac{\sqrt{n} + \sqrt{L}}{(1 - \rho)nL}}.$$

The RHS of the bound above, when $n = L$, is of $O\left(\sqrt{\frac{1}{n(1-\rho)}}\right)$, where $(1 - \rho)$ is the fraction of observed 1's in $\tilde{\mathsf{Y}}$. Naturally, as $\rho$ is large, we need more samples to achieve similar rates as in the other settings.

**Remark 6.** *This PU learning result is particularly very useful in extreme classification setting [Bhatia et al., 2015a, Prabhu and Varma, 2014]; where there are too many labels and is unrealistic to get feedback on every label, but possible to obtain a small subset of relevant labels for instances. Furthermore, the above result serves to attest to the utility of our framework.*

## 5 Experiments

We focus on multi-label datasets for experimental study. The goal is to show that the convergence happens as suggested by the theory, and that the proposed algorithm performs well on real-world datasets. To solve (6), we use an alternating minimization procedure by forming $\mathsf{W} = \mathsf{W}_1 \mathsf{W}_2^T$, such that $\mathsf{W}_1 \in \mathbb{R}^{d \times k}$ and $\mathsf{W}_2 \in \mathbb{R}^{L \times k}$, where $k$, the rank of $\mathsf{W}$, is an input parameter.

### 5.1 Synthetic data

We generate multi-label data as follows. We fix $n = 1000, L = 100$ and $d = 10$. First, we form $\mathsf{X} \in \mathbb{R}^{n \times d}$ using iid samples from multi-variate Gaussian $\mathcal{N}(0, I)$. Then, we generate $\mathsf{W}^*$ of rank 5. The label matrix $\mathsf{Y}$ is obtained by thresholding $\mathsf{X}\mathsf{W}^*$ at $\theta^* = 0$, i.e. $\mathsf{Y}_{ij} = \text{sign}(\langle \mathbf{x}_i, \mathbf{w}_j^* \rangle)$. In this noise-free setting, we expect that our algorithm would recover both $\mathsf{W}^*$ and $\theta^*$ accurately as it sees more and more observations. The results for maximizing micro $F_1$ and accuracy metrics are presented

in Figure 1. As the sampling ratio $\frac{|\Omega|}{nL}$ increases, we observe that the proposed estimator achieves optimal performance in both the cases. Furthermore, even when only 10% of the observations is revealed, we observe that the proposed method achieves very high $F_1$ as well as accuracy values, compared to learning the columns of $\mathbf{W}^*$ independently via the plugin estimator method proposed by [Koyejo et al., 2015] (followed by learning a threshold).

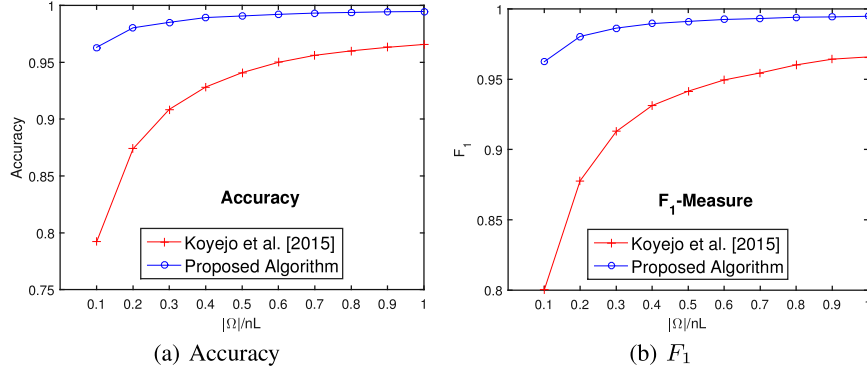

(a) Accuracy                          (b) $F_1$

Figure 1: Convergence of the methods for Accuracy and micro-$F_1$ metrics on synthetic data.

| DATASET | Koyejo et al. [2015] micro $F_1$ | Algorithm 1 micro $F_1$ | Koyejo et al. [2015] Accuracy | Algorithm 1 Accuracy |
|---|---|---|---|---|
| CAL500 | $0.4267 \pm 0.0016$ | $0.3855 \pm 0.0005$ | $0.8541 \pm 0.0034$ | $0.8493 \pm 0.0002$ |
| AUTOFOOD | $0.4897 \pm 0.0103$ | $0.5597 \pm 0.0047$ | $0.9307 \pm 0.0064$ | $0.9345 \pm 0.0043$ |
| BIBTEX | $0.2641 \pm 0.0251$ | $0.2398 \pm 0.0133$ | $0.9849 \pm 0.0016$ | $0.9856 \pm 0.0003$ |
| COMPPHYS | $0.2463 \pm 0.0315$ | $0.3510 \pm 0.0293$ | $0.9448 \pm 0.0011$ | $0.9466 \pm 0.0012$ |
| COREL5K | $0.1552 \pm 0.0116$ | $0.1642 \pm 0.0001$ | $0.9906 \pm 0.0000$ | $0.9906 \pm 0.0000$ |

Table 1: Comparison of proposed algorithm and plugin-estimator method of [Koyejo et al., 2015] on multi-label micro $F_1$ and Hamming (i.e. Accuracy) metrics. Reported values correspond to *micro-averaged* metric computed on test data. In all the cases, $\frac{|\Omega|}{nL}$ was fixed to 20% for training. The rank of W was set to $0.4L$ for Algorithm 1. We observe that the proposed algorithm which captures label correlations is competitive across datasets.

## 5.2 Real-world data

We consider five real-world multi-label datasets widely used as benchmarks [Bhatia et al., 2015a, Yu et al., 2014]. (i) CAL500: a music dataset with 400 training and 100 test instances, $L = 174$, $d = 68$, (ii) COREL5K: an image dataset with 4500 training and 500 test instances, $L = 374$, $d = 499$, (iii) BIBTEX: a text dataset with 4,880 training and 2,515 test instances, $L = 159$, $d = 1,836$, (iv) COMPPHYS dataset with 161 training and 40 test instances, $L = 208$, $d = 33,284$, and (v) AUTOFOOD dataset with 4,880 training and 2,515 test instances, $L = 162$, $d = 1,836$.

We set the rank $k$ of W to $0.4L$ for all the datasets in our method, and set $\frac{|\Omega|}{nL} = 20\%$ to train the models in each method. The results are presented in Table 1. We observe that the proposed method is competitive in all the datasets, and achieves better micro-$F_1$ and accuracy values, with a small value of rank $0.4L$. We note that the label matrices of most of the datasets are very sparse (for instance, less than 8.5% of the test data are positive labels in AUTOFOOD), which explains high accuracy and low $F_1$ values. The learned model is much more compact than that of [Koyejo et al., 2015] ($k(d + L)$ vs $dL$ parameters). While in theory our bounds hold for the case $L \geq d$ (Theorem 3), some of the datasets considered here have $d \gg L$ and yet the performance is competitive.

## 6 Conclusions

We presented a framework for optimizing general performance metrics applicable to multi-label as well as collaborative filtering settings. Our work complements recent results in this direction: on the theoretical front, we derive strong regret bounds for practically used metrics like $F$-measure, and on the algorithmic front, we provide simple and efficient procedure that works well in practice.

## Footnotes

[1]The definitions in [Koyejo et al., 2015] do not include general sampling distribution $\pi$, but the results can be generalized in a straight-forward manner.

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
