[Supplementary Material]

# A Appendix: Proofs

## A.1 Proof of Theorem 1

Proof technique is based on [Kotłowski and Dembczyński, 2015], where they derive a similar bound in the binary classification setting. We first relate the $\Psi$-regret to weighted 0-1 loss regret. Define the $\alpha$-weighted 0-1 loss $\ell_\alpha : \mathbb{R} \times \mathbb{R} \to [0,1]$ as:

$$\ell_\alpha(\hat{y}, y) = \alpha[\![y = 0]\!][\![\hat{y} = 1]\!] + (1 - \alpha)[\![y = 1]\!][\![\hat{y} = 0]\!],$$

Let $\widehat{\mathsf{Y}} = f(\mathsf{X})$ for some function $f$. The $\ell_\alpha$-risk of $f$ with respect to the underlying distribution over $\mathsf{X}, \mathsf{Y}$ and $\Omega$ is defined as:

$$\mathrm{Risk}_\alpha(\widehat{\mathsf{Y}}, \mathsf{Y}) = \mathbb{E}[\ell_\alpha(\widehat{\mathsf{Y}}_{ij}, \mathsf{Y}_{ij})] = \alpha\mathrm{FP}(\widehat{\mathsf{Y}}, \mathsf{Y}) + (1 - \alpha)\mathrm{FN}(\widehat{\mathsf{Y}}, \mathsf{Y}).$$

Define the Bayes optimal corresponding to the above risk: $f_\alpha^*(\mathsf{X}) = \arg\min_f \mathrm{Risk}_\alpha(f(\mathsf{X}), \mathsf{Y})$. Let $\mathrm{Risk}_\alpha^* := \mathrm{Risk}(f_\alpha^*(\mathsf{X}))$. The $\ell_\alpha$-regret of $f$ is defined as:

$$\mathrm{Reg}_\alpha(f(\mathsf{X})) := \mathrm{Risk}_\alpha(f(\mathsf{X})) - \mathrm{Risk}_\alpha^*.$$

**Lemma 2.** *Let $\Psi$ be a linear-fractional performance metric as defined in* (3), (4) *or* (5). *Then for $\alpha \in (0,1)$ defined as:*

$$\alpha = \frac{\Psi^* c_2 - c_1}{\Psi^* c_2 - c_1 + \Psi^* d_2 - d_1}, \tag{9}$$

*where $c_1, d_1, c_2, d_2$ are constants that depend on $\Psi$, there exists some constant $C > 0$ such that, for any $f$:*

$$\Psi^* - \Psi(f(\mathsf{X}), \mathsf{Y}) \leq C(\mathrm{Risk}_\alpha(f(\mathsf{X}), \mathsf{Y}) - \mathrm{Risk}_\alpha^*). \tag{10}$$

Let $\ell : \{0,1\} \times \mathbb{R} \to \mathbb{R}_+$ be a $\lambda'$-strongly proper composite loss [Agarwal, 2014], such as the squared loss or the logistic. Given real-valued predictions $\mathsf{Z} \in \mathbb{R}^{n \times L}$, we now argue that there exists a thresholding $\mathrm{Thr}_{\theta^*}(\mathsf{Z}) \in \{0,1\}^{n \times L}$ such that $\mathrm{Risk}_\alpha(\mathrm{Thr}_{\theta^*}(\mathsf{Z}), \mathsf{Y})$ is bounded by the $\ell$-regret of a strongly proper loss $\ell$ (where Thr operator is defined as in Step 2 of Algorithm 1).

**Lemma 3.** *Let $\ell$ be a $\lambda$-strongly proper loss function, and $\alpha$ be defined as in* (9). *Then, there exists $\theta^*$ s.t.*

$$\mathrm{Reg}_\alpha(Thr_{\theta^*}(\mathsf{Z})) \leq \sqrt{\frac{2}{\lambda}}\sqrt{\mathrm{Reg}_\ell(\mathsf{Z})}.$$

Finally, we show that estimating $\hat{\theta}$ from training samples (Step 3 of Algorithm 1) is sufficient for bounding the $\Psi$-regret.

**Lemma 4.** *We have:*

$$\max_\theta \widehat{\Psi}(\mathrm{Thr}_\theta(\mathsf{Z}), \mathsf{Y}) \geq \widehat{\Psi}(\mathrm{Thr}_{\theta^*}(\mathsf{Z}), \mathsf{Y}),$$

*and*

$$\max_\theta \widehat{\Psi}(\mathrm{Thr}_\theta(\mathsf{Z}_\Omega), \mathsf{Y}_\Omega) \geq \max_\theta \Psi(\mathrm{Thr}_\theta(\mathsf{Z}), \mathsf{Y}) - O\left(\frac{1}{\sqrt{|\Omega|}}\right).$$

The proof of the Theorem is complete by chaining the above three Lemmas. □

**Remark 7.** *When $\Psi^*$ is known (in the noise-free or realizable setting, $\Psi^*$ is the maximum possible value of $\Psi$), we can get a closed form for $\theta^*$, which is $\theta^* = \xi(\alpha)$ where $\xi$ is the link function corresponding to the proper loss $\ell$.*

### A.1.1 Proof of Lemma 2

Let $\widehat{\mathsf{Y}} = f(\mathsf{X})$. Consider the metric $\Psi$ from family (3) for the moment. Define $A(\widehat{\mathsf{Y}}) = a_0 + a_{11}\mathrm{TP} + a_{01}\mathrm{FP} + a_{10}\mathrm{FN} + a_{00}\mathrm{TN} := c_1\mathrm{FP} + d_1\mathrm{FN} + e_1$ and $B(\widehat{\mathsf{Y}}) = b_0 + b_{11}\mathrm{TP} + b_{01}\mathrm{FP} + b_{10}\mathrm{FN} + b_{00}\mathrm{TN} :=$

$c_2\text{FP}+d_2\text{FN}+e_2$ (for constants $c_1, c_2, d_1, d_2, e_1, e_2$ suitably defined), so that $\Psi(\widehat{\mathbf{Y}}, \mathbf{Y}) = A(\widehat{\mathbf{Y}})/B(\widehat{\mathbf{Y}})$. Let $f^*$ denote the Bayes optimal attaining $\Psi^* = A^*/B^*$. We have:

$$
\begin{aligned}
\Psi^* - \Psi(\widehat{\mathbf{Y}}, \mathbf{Y}) &= \frac{\Psi^* B(\widehat{\mathbf{Y}}) - A(\widehat{\mathbf{Y}})}{B(\widehat{\mathbf{Y}})} \\
&= \frac{\Psi^* B(\widehat{\mathbf{Y}}) - A(\widehat{\mathbf{Y}}) - (\Psi^* B^* - A^*)}{B(\widehat{\mathbf{Y}})} \\
&= \frac{\Psi^* (B(\widehat{\mathbf{Y}}) - B^*) - (A(\widehat{\mathbf{Y}}) - A^*)}{B(\widehat{\mathbf{Y}})} \\
&= \frac{(\Psi^* c_2 - c_1)(\text{FP}(\widehat{\mathbf{Y}}, \mathbf{Y}) - \text{FP}(f^*(\mathbf{X}), \mathbf{Y})) + (\Psi^* d_2 - d_1)(\text{FN}(\widehat{\mathbf{Y}}, \mathbf{Y}) - \text{FN}(f^*(\mathbf{X}), \mathbf{Y}))}{B(\widehat{\mathbf{Y}})} \\
&\leq \frac{(\Psi^* c_2 - c_1)(\text{FP}(\widehat{\mathbf{Y}}, \mathbf{Y}) - \text{FP}(f^*(\mathbf{X}), \mathbf{Y})) + (\Psi^* d_2 - d_1)(\text{FN}(\widehat{\mathbf{Y}}, \mathbf{Y}) - \text{FN}(f^*(\mathbf{X}), \mathbf{Y}))}{\gamma'} \\
&= C\big(\text{Risk}_\alpha(\widehat{\mathbf{Y}}, \mathbf{Y}) - \text{Risk}_\alpha(f^*(\mathbf{X}), \mathbf{Y})\big).
\end{aligned}
$$

Assuming $(\Psi^* c_2 - c_1) \geq 0$ and $(\Psi^* d_2 - d_1) \geq 0$, the last equality follows by defining:

$$
\alpha = \frac{\Psi^* c_2 - c_1}{\Psi^* c_2 - c_1 + \Psi^* d_2 - d_1}. \tag{11}
$$

and $C = \frac{\Psi^* c_2 - c_1 + \Psi^* d_2 - d_1}{\gamma'}$. The statement of the lemma follows. When $\Psi$ is a metric from family (4), we can apply Proposition 1 of [Koyejo et al., 2015] to see that $\text{TP}_i = \text{TP}$, $\text{FP}_i = \text{FP}$ and so on (as the expectations are defined wrt $\text{TP}_{ij}, \text{FP}_{ij}$), which yields $\Psi^*$ is identical as in the micro-averaging case. So, the same regret bound applies as shown below: Define $A_i = a_0 + a_{11}\text{TP}_i + a_{01}\text{FP}_i + a_{10}\text{FN}_i + a_{00}\text{TN}_i = c_1\text{FP}_i + d_1\text{FN}_i + e_1$ and $B_i$ similarly. As before, let $\Psi^* = A^*/B^*$. So when $\Psi$ is of the form (4),

$$
\begin{aligned}
\Psi^* - \Psi(\widehat{\mathbf{Y}}, \mathbf{Y}) &= \frac{1}{n}\sum_{i=1}^n \frac{\Psi^* B_i(\widehat{\mathbf{Y}}) - A_i(\widehat{\mathbf{Y}})}{B_i(\widehat{\mathbf{Y}})} \\
&= \frac{1}{n}\sum_{i=1}^n \frac{\Psi^* B_i(\widehat{\mathbf{Y}}) - A_i(\widehat{\mathbf{Y}}) - (\Psi^* B^* - A^*)}{B_i(\widehat{\mathbf{Y}})} \\
&= \frac{1}{n}\sum_{i=1}^n \frac{\Psi^* (B_i(\widehat{\mathbf{Y}}) - B^*) - (A_i(\widehat{\mathbf{Y}}) - A^*)}{B_i(\widehat{\mathbf{Y}})} \\
&= \frac{1}{n}\sum_{i=1}^n \frac{(\Psi^* c_2 - c_1)(\text{FP}_i(\widehat{\mathbf{Y}}, \mathbf{Y}) - \text{FP}(f^*(\mathbf{X}), \mathbf{Y})) + (\Psi^* d_2 - d_1)(\text{FN}_i(\widehat{\mathbf{Y}}, \mathbf{Y}) - \text{FN}(f^*(\mathbf{X}), \mathbf{Y}))}{B_i(\widehat{\mathbf{Y}})} \\
&= \frac{1}{n}\sum_{i=1}^n \frac{(\Psi^* c_2 - c_1)(\text{FP}(\widehat{\mathbf{Y}}, \mathbf{Y}) - \text{FP}(f^*(\mathbf{X}), \mathbf{Y})) + (\Psi^* d_2 - d_1)(\text{FN}(\widehat{\mathbf{Y}}, \mathbf{Y}) - \text{FN}(f^*(\mathbf{X}), \mathbf{Y}))}{B_i(\widehat{\mathbf{Y}})} \\
&\leq \frac{(\Psi^* c_2 - c_1)(\text{FP}(\widehat{\mathbf{Y}}, \mathbf{Y}) - \text{FP}(f^*(\mathbf{X}), \mathbf{Y})) + (\Psi^* d_2 - d_1)(\text{FN}(\widehat{\mathbf{Y}}, \mathbf{Y}) - \text{FN}(f^*(\mathbf{X}), \mathbf{Y}))}{\gamma'} \\
&= C\big(\text{Risk}_\alpha(\widehat{\mathbf{Y}}, \mathbf{Y}) - \text{Risk}_\alpha(f^*(\mathbf{X}), \mathbf{Y})\big).
\end{aligned}
$$

which is identical to the bound for family (3). It is easy to see that (5) also admits the above bound. Therefore, relation (10) holds for all definitions of $\Psi$, with the same $\alpha$.

### A.1.2 Proof of Lemma 3

Let $\mathbf{Y}, \widehat{\mathbf{Y}} \in \{0,1\}^{n \times L}$. Note that for any $\ell$, $\text{Risk}_\ell(f)$ is defined as:

$$
\text{Risk}_\ell(f) = \mathbb{E}[\ell(\widehat{\mathbf{Y}}_{ij}, \mathbf{Y}_{ij})] = \mathbb{E}_{\mathbf{X} \sim \mathbb{P}_\mathbf{X}^n} \mathbb{E}_{(i,j) \sim \pi} \mathbb{E}_{\mathbf{Y}_{ij} \sim \mathbb{P}(.|\mathbf{x}_i)} \ell(\widehat{\mathbf{Y}}_{ij}, \mathbf{Y}_{ij}),
$$

where $\pi$ denotes the sampling distribution over $(i,j)$ pairs. Fix instance $i$ and label $j$. Let $\eta_{ij}$ denote the conditional probability of label $j$ of instance $i$ being 1, i.e. $\eta_{ij} = \mathbb{P}(\mathbf{Y}_{ij} = 1|\mathbf{x}_i)$. For convenience,

denote $\eta_{ij}$ simply by $\eta$. Given $\eta \in [0,1]$, and $\hat{y} \in \{0,1\}$, consider the conditional $\ell_\alpha$-risk of $\hat{y}$:

$$L_\alpha(\eta, \hat{y}) = \alpha(1-\eta)[[\hat{y}=1]] + (1-\alpha)\eta[[\hat{y}=0]],$$

and the corresponding conditional $\ell_\alpha$ regret of $\hat{y}$:

$$\text{Reg}_\alpha^L(\eta, \hat{y}) = L_\alpha(\eta, \hat{y}) - \min_{\hat{y}} L_\alpha(\eta, \hat{y}),$$

where we have: $\arg\min_{\hat{y}} L_\alpha(\eta, \hat{y}) = [[\eta - \alpha]]$.
More generally, for a loss $\ell$, and a number $\hat{z}$, we have:

$$L_\ell(\eta, \hat{z}) = \ell(\hat{z}, 1)\eta + \ell(\hat{z}, 0)(1-\eta),$$

and

$$\text{Reg}_\ell^L(\eta, \hat{z}) = L_\ell(\eta, \hat{z}) - \min_{\hat{z}} L_\ell(\eta, \hat{z}).$$

Now, observe that:

$$\text{Risk}_\alpha(\widehat{\mathsf{Y}}, \mathsf{Y}) = \mathbb{E}_{\mathsf{X} \sim \mathbb{P}_\mathsf{X}^n} \mathbb{E}_{(i,j) \sim \pi} L_\alpha(\eta_{ij}, \widehat{\mathsf{Y}}_{ij}),$$

and

$$\text{Reg}_\alpha(\widehat{\mathsf{Y}}, \mathsf{Y}) = \mathbb{E}_{\mathsf{X} \sim \mathbb{P}_\mathsf{X}^n} \mathbb{E}_{(i,j) \sim \pi} \text{Reg}_\alpha^L(\eta_{ij}, \widehat{\mathsf{Y}}_{ij}),$$

where the last equality follows from the fact that the Bayes optimal $f_\alpha^*$ of the $\ell_\alpha$-risk minimizes the conditional $L_\alpha(\eta_{ij}, .)$ risk for each $(i,j)$. Let $\mathsf{Z} = f(\mathsf{X}) \in \mathbb{R}^{n \times L}$ denote real-valued predictions obtained using some function $f$. Using the same arguments as by Kotłowski and Dembczyński [2015], we can show that, by setting threshold $\theta^* = \xi(\alpha)$, where $\xi$ is the monotonic link function corresponding to $\lambda'$-strongly proper loss $\ell$, and $\alpha$ is defined as in (9), the conditional $\ell_\alpha$ regret of $\widehat{\mathsf{Y}}_{ij} = [[\mathsf{Z}_{ij} \geq \theta^*]]$ for a fixed $(i,j)$ can be bounded as:

$$\text{Reg}_\alpha^L(\eta_{ij}, \widehat{\mathsf{Y}}_{ij}) \leq \sqrt{\frac{2}{\lambda'}} \sqrt{\text{Reg}_\ell^L(\eta_{ij}, \mathsf{Z}_{ij})},$$

Taking expectation wrt sampling distribution $\pi$ and the distribution over instances $\mathbb{P}_\mathsf{X}^n$ on both the sides of the above inequality, and applying Jensen's inequality, the statement of the Lemma follows.

### A.1.3 Proof of Lemma 4

The first part of the lemma is trivially true. For the second part, we can apply the same arguments as in Lemma 9 of Koyejo et al. [2014].

### A.2 Proof of Theorem 2

The following theorem bounds the error of the estimator $\hat{\mathsf{W}} \in \mathbb{R}^{n \times L}$ in this model, via the result by Lafond [2015].

**Theorem 5** ( Lafond [2015]). *Assume $\pi$ is uniform, and consider the 1-bit matrix completion sampling model (2). Let $\hat{W}$ be the solution to the trace-norm regularized optimization problem (6) using logistic loss for $\ell$ (with input $\mathsf{X}$ assumed to be identity matrix of size $n$), number of observations $|\Omega| \geq \log(n+L)\min(n,L)\max(c_\gamma' \log^2(c_\gamma'' \sqrt{\min(n,L)}), 1/9)$, and setting the regularization parameter $\lambda = 2c_\gamma \sqrt{\frac{2\log(n+L)}{\min(n,L)|\Omega|}}$. Then, with probability at least $1 - 3(n+L)^{-1}$, the following holds:*

$$\frac{\|\hat{W} - W^*\|_F^2}{nL} \leq \tilde{C} \max\left(\frac{\max(n,L)\, \mathit{rank}(W^*)\log(n+L)}{|\Omega|}\left(\sigma_\gamma^2 + 1\right), \gamma^2 \sqrt{\frac{\log(n+L)}{|\Omega|}}\right),$$

*where $\tilde{C}, c_\gamma, c_\gamma', c_\gamma'', \sigma_\gamma$ are numerical constants and $\gamma = \max_{ij} |W_{ij}^*|$.*

The above theorem can be extended to general distributions $\pi$ satisfying Assumption 2. See Lafond [2015] for more details. Now, we use the fact that $\ell$ is 1-Lipschitz (say, by choosing logistic loss), and bound $\mathbb{E}[\ell(\hat{\mathsf{W}}_{ij}, \mathsf{Y}_{ij}) - \ell(\mathsf{W}_{ij}^*, \mathsf{Y}_{ij})] \leq \frac{1}{nL}\sum_{ij} |\hat{\mathsf{W}}_{ij} - \mathsf{W}_{ij}^*|$. Observing that $\|\hat{\mathsf{W}} - \mathsf{W}^*\|_1 \leq \sqrt{nL}\|\hat{\mathsf{W}} - \mathsf{W}^*\|_F$, and combining with the bound in Theorem 5, the proof is complete.

### A.3 Weakness of using Lafond [2015] for Multi-label Learning

In the multi-label learning model (1), one could hope to directly apply the analysis of Lafond [2015] for recovering $\mathsf{XW}^* \in \mathbb{R}^{n \times L}$, and in turn, $\mathsf{W}^* \in \mathbb{R}^{d \times L}$. In lieu of problem (6), we would then solve the optimization problem in Lafond [2015]:

$$\hat{\mathsf{W}} = \arg \min_{\mathsf{W}:\|\mathsf{XW}\|_\infty \leq \gamma} \frac{1}{|\Omega|} \sum_{(i,j) \in \Omega} \ell(\langle \mathbf{x}_i, \mathbf{w}_j \rangle, \mathsf{Y}_{ij}) + \lambda \|\mathsf{XW}\|_* \tag{12}$$

Note that the only difference is how the trace-norm regularization is performed: $\|\mathsf{XW}\|_*$ versus our proposed $\|\mathsf{W}\|_*$ in Algorithm 1. The following corollary of Theorem 5 provides a bound for the recovery error of $\hat{\mathsf{W}}$.

**Corollary 2.** *Assume 1, $\pi$ is uniform, and consider the sampling model* (1). *Let $\hat{W}$ be the solution to the trace-norm regularized optimization problem* (12) *using logistic loss for $\ell$, number of observations $|\Omega| \geq \log(n+L) \min(n,L) \max(c'_\gamma \log^2(c''_\gamma \sqrt{\min(n,L)}), 1/9)$, and setting the regularization parameter $\lambda = 2c_\gamma \sqrt{\frac{2 \log(n+L)}{\min(n,L)|\Omega|}}$. Then, with probability at least $1 - 3(n + L)^{-1}$, the following holds:*

$$\frac{\|\hat{W} - W^*\|_F^2}{dL} \leq \frac{\tilde{C}}{d} \max \left( \frac{\max(n,L) \, \mathit{rank}(W^*) \log(n+L)}{|\Omega|} \left( \sigma_\gamma^2 + 1 \right), \gamma^2 \sqrt{\frac{\log(n+L)}{|\Omega|}} \right),$$

*where $\tilde{C}, c_\gamma, c'_\gamma, c''_\gamma, \sigma_\gamma$ are numerical constants and $\gamma = \max_{ij} |(\mathit{XW}^*)_{ij}|$.*

*Proof.* In the multilabel setting, Theorem 5 bounds $\|\mathsf{XW} - \mathsf{XW}^*\|_F^2 / nL$, which in turn can be lowerbounded using Lemma 6 and then Lemma 7. Introducing $(1/d)$ on both sides of the resulting inequality gives the average error stated in the corollary. $\square$

When $n \geq L$ and $|\Omega| = O(n)$, which is quite common in multi-label scenario, the above bound suggests that $\hat{\mathsf{W}}$ from (12) is not even a consistent estimator, even when $\pi$ is uniform.

### A.4 Proof of Theorem 3

The statement is a corollary of the more general Theorem 8, proved in Appendix B. We can compute the constants for the logistic loss as: $\bar{\sigma}_\gamma \leq 1$ and $\underline{\sigma}_\gamma \geq \frac{(1+e^\gamma)^2}{e^{-\gamma}}$, over the domain $[-\gamma, \gamma]$.

### A.5 Proof of Theorem 4

The following result by [Hsieh et al., 2015] gives recovery bound for the resulting estimator $\hat{\mathsf{W}}$, as described in the text (Section 4.2.3).

**Theorem 6** ([Hsieh et al., 2015]). *With probability at least $1 - 2(n + L)^{-1}$,*

$$\frac{\|\hat{W} - W^*\|_F^2}{nL} \leq 6 \frac{\sqrt{\log(n+L)}}{\sqrt{nL}(1-\rho)} + 2C \cdot t \frac{\sqrt{n} + \sqrt{L}}{(1-\rho)nL},$$

*where $C$ is absolute constant and $\|\mathsf{W}^*\|_* \leq t$.* The proof is complete by using the same argument for 1-Lipschitz $\ell$ as in the proof of Theorem 2.

# B    Appendix B: Sampling from Exponential Distribution

We now consider the generalized matrix completion problem when the values are sampled iid from an exponential distribution parameterized by the input features $\mathbf{x} \in \mathbb{R}^d$. This setting extends that of Lafond [2015]. Let $y_{ij} \in \mathbb{R}$ denote a random sample corresponding to the user $i$ and label $j$, which is distributed as:

$$y_{ij}|\mathbf{x}_i, \mathbf{w}_j \sim \exp_{h,G}(\mathbf{x}_i, \mathbf{w}_j) := h(y_{ij}) \exp\left(\langle \mathbf{x}_i, \mathbf{w}_j \rangle y_{ij} - G(\langle \mathbf{x}_i, \mathbf{w}_j \rangle)\right). \tag{13}$$

where $\langle \mathbf{x}_i, \mathbf{w}_j \rangle$, $i = 1, 2, \ldots, n$ and $j = 1, 2, \ldots, L$ are the canonical parameters, $h$ and $G$ are the base measure and log-partition functions associated with this canonical representation.

Let $\mathbf{W}^* \in \mathbb{R}^{d \times L}$ denote the ground-truth parameter matrix with $\mathbf{w}_j$'s as columns. Similarly, let $\mathsf{Y} \in \mathbb{R}^{n \times L}$ (with entries $y_{ij}$) denote a random sample from $\mathsf{X}\mathbf{W}^*$. As in the standard matrix completion setting, we only observe values of $\mathsf{Y}$ corresponding to a set of indices $\Omega$ sampled iid from a fixed distribution $\pi$.

**Notation.**    With a slight abuse, we will continue to use $\langle ., . \rangle$ when the arguments are matrices, instead of the **trace** operator, i.e. for matrices $A$ and $B$ of appropriate dimensions, $\langle A, B \rangle := \mathbf{trace}(A^T B)$. Let $\|A\|_\infty = \max_{ij} |A_{ij}|$, $\|A\|_F = \sqrt{\sum_{ij} A_{ij}^2}$, $\|A\|_*$ denote the trace norm (sum of singular values of $A$), $\sigma_{\max}(A) = \|A\|_2$ denote the operator norm (maximum singular value) of $A$, and $\sigma_{\min}(A)$ denote its smallest singular value.

**Maximum Log-likelihood Estimator.**

We consider the negative log-likelihood of the observations, given by:

$$\Phi_Y(\mathsf{X}, \mathbf{W}) = -\frac{1}{|\Omega|} \sum_{(i,j) \in |\Omega|} y_{ij} \langle \mathbf{x}_i, \mathbf{w}_j \rangle - G(\langle \mathbf{x}_i, \mathbf{w}_j \rangle).$$

Constrained ML estimator is obtained as:

$$\hat{\mathbf{W}} := \arg \min_{\mathbf{W}: \|\mathsf{X}\mathbf{W}\|_\infty \leq \gamma} \Phi_Y^\lambda(\mathsf{X}, \mathbf{W}) := \Phi_Y(\mathsf{X}, \mathbf{W}) + \lambda \|\mathbf{W}\|_* \tag{14}$$

**Assumption 3.**        *1. The function $G(x)$ is twice differentiable and strongly convex on $[-\gamma, \gamma]$, such that there exists constants $\bar{\sigma}_\gamma > 0$ and $\underline{\sigma}_\gamma > 0$ satisfying:*

$$\underline{\sigma}_\gamma^2 \leq G''(x) \leq \bar{\sigma}_\gamma^2,$$

*for any $x \in [-\gamma, \gamma]$.*

*2. There exists a constant $\delta_\gamma > 0$ such that for all $x \in [-\gamma, \gamma]$ and $y \sim \exp_{h,G}(x)$:*

$$\mathbb{E}_{y \sim \mathbb{P}(.|x)}\left[\exp\left(\frac{|y - G'(x)|}{\delta_\gamma}\right)\right] \leq e.$$

**Definition 1.** *Given convex function $G(x)$ define the Bregman divergence between two scalars $x, x' \in \mathbb{R}$ as:*

$$d_G(x, x') = G(x) - G(x') - G'(x')(x - x'). \tag{15}$$

**Remark 8.** *Under Assumption 3.1, for any $x, x' \in [-\gamma, \gamma]$, the Bregman divergence $G$ satisfies:*

$$\underline{\sigma}_\gamma^2(x - x')^2 \leq 2d_G(x, x') \leq \bar{\sigma}_\gamma^2(x - x')^2. \tag{16}$$

Let $E_{ij} \in \mathbb{R}^{n \times L}$ denote the indicator matrix with zeros everywhere except at $(i, j)$ where it is 1. For $(\epsilon_{ij})_{ij=1}^{|\Omega|}$ a Rademacher sequence independent from $(\Omega, Y_\Omega)$, define:

$$\Sigma_R := \frac{1}{|\Omega|} \sum_{(i,j) \in \Omega} \epsilon_{ij} E_{ij}. \tag{17}$$

**Theorem 7.** *Assume 3.1, 2.1, $\|XW^*\|_\infty \le \gamma$, $\sigma_{\min}(X) > 0$ and $2\|X^T \nabla \Phi_Y(X, W^*)\|_2 \le \lambda$. Then, with probability at least $1 - 2(n + L)^{-1}$, the following holds:*

$$\frac{\|\hat{W} - W^*\|_F^2}{dL} \le \frac{C\mu^2 n}{\sigma_{min}^2(X) \cdot d} \max\left( L \, rank(W^*)\left(\frac{\lambda^2}{\underline{\sigma}_\gamma^4}\frac{n}{\sigma_{\min}^2(X)} + d\big(\mathbb{E}\|\Sigma_R\|_2\big)^2\right), \frac{\gamma^2}{\mu}\sqrt{\frac{\log(n+L)}{|\Omega|}}\right),$$

*where $C$ is a numerical constant and $\Sigma_R$ is defined as in (17).*

*Proof.* The proof closely follows that of Theorem 5 of Lafond [2015]. As $\hat{W}$ is the minimizer of (14), we have:
$$\Phi_Y^\lambda(X, \hat{W}) - \Phi_Y^\lambda(X, W^*) \le 0$$
It follows that:
$$\lambda(\|\hat{W}\|_* - \|W^*\|_*) + \frac{1}{|\Omega|}\sum_{(i,j)\in\Omega} y_{ij}\langle \mathbf{x}_i, \mathbf{w}_j^* - \hat{\mathbf{w}}_j\rangle + G(\langle \mathbf{x}_i, \hat{\mathbf{w}}_j\rangle) - G(\langle \mathbf{x}_i, \mathbf{w}_j^*\rangle) \le 0$$

Using the fact that the gradient matrix:
$$\nabla \Phi_Y(X, W^*) := \nabla_{XW^*}\Phi_Y(X, W^*) = -\frac{1}{|\Omega|}\sum_{(i,j)\in\Omega}\big(y_{ij} - G'(\langle \mathbf{x}_i, \mathbf{w}_j^*\rangle)\big)E_{ij} \qquad (18)$$

(where $E_{ij}$ are the indicator matrices defined earlier) in the above inequality, we have:
$$\lambda(\|\hat{W}\|_* - \|W^*\|_*) + \left\langle \nabla \Phi_Y(X, W^*), X(W^* - \hat{W})\right\rangle +$$

$$\frac{1}{|\Omega|}\sum_{(i,j)\in\Omega} G(\langle \mathbf{x}_i, \hat{\mathbf{w}}_j\rangle) - G(\langle \mathbf{x}_i, \mathbf{w}_j^*\rangle) - G'(\langle \mathbf{x}_i, \mathbf{w}_j^*\rangle)(\langle \mathbf{x}_i, \hat{\mathbf{w}}_j - \mathbf{w}_j^*\rangle) \le 0.$$

Using the definition of the divergence (15), and the fact that $\left\langle \nabla \Phi_Y(X, W^*), X(W^* - \hat{W})\right\rangle = \left\langle X^T \nabla \Phi_Y(X, W^*), W^* - \hat{W}\right\rangle$ it follows that:
$$D_G^\Omega(X\hat{W}, XW^*) := \frac{1}{|\Omega|}\sum_{(i,j)\in\Omega} d_G(\langle \mathbf{x}_i, \hat{\mathbf{w}}_j\rangle, \langle \mathbf{x}_i, \mathbf{w}_j^*\rangle) \le \lambda(\|W^*\|_* - \|\hat{W}\|_*) - \left\langle X^T\nabla\Phi_Y(X, W^*), W^* - \hat{W}\right\rangle$$

The first term in the RHS of above inequality can be bounded first using Lemma 16-(iii) of Lafond [2015]. The second term can be bounded using the trace inequality (that uses the duality between $\|.\|_*$ and $\|.\|_2$) and the assumption on $\lambda$ stated in the Theorem. We get:
$$D_G^\Omega(X\hat{W}, XW^*) \le \lambda(\|\mathcal{P}_{W^*}(\hat{W} - W^*)\|_* + \frac{1}{2}\|\hat{W} - W^*\|_*).$$

To bound the first term in the above equation, we can apply Lemma 16-(ii) of Lafond [2015]. Lemma 5 gives a bound for the second term. Together we have:
$$D_G^\Omega(X\hat{W}, XW^*) \le 3\lambda\sqrt{2\,rank(W^*)}\|\hat{W} - W^*\|_F. \qquad (19)$$

By strong convexity of $G$ (Assumption 3.1), we have:
$$\Delta_Y^2(X\hat{W}, XW^*) := \frac{1}{|\Omega|}\sum_{(i,j)\in\Omega}(\langle \mathbf{x}_i, \hat{\mathbf{w}}_j - \mathbf{w}_j^*\rangle)^2 \le \frac{2}{\underline{\sigma}_\gamma^2} D_G^\Omega(X\hat{W}, XW^*). \qquad (20)$$

Now, we will get a lower bound for $\Delta_Y^2(X\hat{W}, XW^*)$. To do so, let us define $\beta := 8e\gamma^2\sqrt{\log(n+L)/|\Omega|}$ and distinguish the two following cases:

**Case 1** If $\mathbb{E}[(\langle \mathbf{x}_i, \hat{\mathbf{w}}_j - \mathbf{w}_j^*\rangle)^2] \le \beta$, where $\mathbb{E}$ is defined wrt the sampling distribution as in Assumption 2, then Lemma 18 of Lafond [2015] yields,
$$\frac{\|X\hat{W} - XW^*\|_F^2}{nL} \le \mu\beta. \qquad (21)$$

**Case 2** If $\mathbb{E}[(\langle \mathbf{x}_i, \hat{\mathbf{w}}_j - \mathbf{w}_j^* \rangle)^2] > \beta$, consider $\hat{\mathbf{W}} \in \mathcal{C}(\beta, 32\mu dL\, \mathsf{rank}(\mathbf{W}^*))$, where $\mathcal{C}(.,.)$ is defined as:

$$\mathcal{C}(\beta, r) = \left\{ \mathbf{W} \in \mathbb{R}^{d \times L} \mid \|\mathbf{W}^* - \hat{\mathbf{W}}\|_* \leq \sqrt{r\mathbb{E}[\Delta_{\mathsf{Y}}^2(\mathbf{XW}, \mathbf{XW}^*)]}; \mathbb{E}[\Delta_{\mathsf{Y}}^2(\mathbf{XW}, \mathbf{XW}^*)] > \beta \right\}. \quad (22)$$

Then, from Lemma 19 of Lafond [2015], it holds with probability at least $1 - 2(n+L)^{-1}$ that

$$\Delta_{\mathsf{Y}}^2(\mathbf{X\hat{W}}, \mathbf{XW}^*) \geq \frac{1}{2}\mathbb{E}[\Delta_{\mathsf{Y}}^2(\mathbf{X\hat{W}}, \mathbf{XW}^*)] - 512e(\mathbb{E}[\|\Sigma_R\|_2])^2 \mu dL\, \mathsf{rank}(\mathbf{W}^*). \quad (23)$$

Combining the above inequality with (20), (19) and Lemma 18 of Lafond [2015] yields:

$$\frac{\|\mathbf{X\hat{W}} - \mathbf{XW}^*\|_F^2}{2\mu nL} - 512e(\mathbb{E}[\|\Sigma_R\|_2])^2 \mu dL\, \mathsf{rank}(\mathbf{W}^*) \leq \frac{6\lambda}{\underline{\sigma}_\gamma^2}\sqrt{2\,\mathsf{rank}(\mathbf{W}^*)}\|\hat{\mathbf{W}} - \mathbf{W}^*\|_F.$$

We can use Lemma (6) to bound the first term from below. Applying the identity $ab \leq (a^2 + b^2)/4$, multiplying both sides of the inequality by $1/d$, rearranging and combining with (21), the proof is complete. $\qquad\square$

**Theorem 8.** *Assume 1, 2, 3. Choose, $n \geq C'\,.\,d$, $L \geq d$, $|\Omega| \geq L + d$ and $\lambda = \frac{2c\bar{\sigma}_\gamma}{\sqrt{|\Omega|}}$. Then, with probability at least $1 - 3(n+L)^{-1} - 2(d+L)^{-1}$, the following holds:*

$$\frac{\|\hat{\mathbf{W}} - W^*\|_F^2}{dL} \leq \frac{C_2\mu^2}{d}\max\left( \frac{L\,\mathsf{rank}(\mathbf{W}^*)\log(n+L)}{|\Omega|}\left(\frac{\bar{\sigma}_\gamma^2}{\underline{\sigma}_\gamma^4} + 1\right), \frac{\gamma^2}{\mu}\sqrt{\frac{\log(n+L)}{|\Omega|}} \right),$$

*where $c, C', C_2$ are numerical constants.*

*Proof.* It suffices to show $2\|\mathbf{X}^T\nabla\Phi(\mathbf{X}, \mathbf{W}^*)\|_2 \leq \lambda$ for chosen $\lambda$ in the statement of the Theorem and a suitable bound for $\mathbb{E}\|\Sigma_R\|_2$ (the result would then follow by applying Theorem 7). The latter term can be readily bounded applying the corresponding arguments in the proof of Theorem 6 of Lafond [2015], which yields:

$$\mathbb{E}\|\Sigma_R\|_2 \leq c^*\sqrt{\frac{2e\log(n+L)}{|\Omega|}\left(\frac{\nu}{\min(n,L)}\right)}, \quad (24)$$

where we use the fact that $\sum_{l=1}^{L}\pi_{k,l} = \frac{\nu}{\min(n,L)}$ (by Assumption 2). where $c^*$ is a numerical constant.

We can apply Lemma 1 to bound $\|\mathbf{X}^T\nabla\Phi(\mathbf{X}, \mathbf{W}^*)\|_2$, with the $\lambda$ chosen in the statement of the Theorem. The proof is complete noting that for the choice of $n$ as in the statement of the Theorem, Lemma 7 implies $\sigma_{\min}^2(X) \geq \underline{C}n$ and that for the choice of $n$ and $L$ as in the statement of the Theorem, $\frac{d}{\min(n,L)} \leq 1$.

$\qquad\square$

**Lemma 5.** *Let $XW, X\tilde{W} \in \mathbb{R}^{n \times L}$ satisfy $\|XW\|_\infty \leq \gamma$ and $\|X\tilde{W}\|_\infty \leq \gamma$. Assume $2\|X^T\nabla\Phi_{\mathsf{Y}}(X, \tilde{W})\|_2 \leq \lambda$, and $\Phi_Y^\lambda(X, W) \leq \Phi_Y^\lambda(X, \tilde{W})$. Then:*
*(i) $\|\mathcal{P}_{\tilde{W}}^\perp(W - \tilde{W})\|_* \leq 3\|\mathcal{P}_{\tilde{W}}(W - \tilde{W})\|_*$,*
*(ii) $\|W - \tilde{W}\|_* \leq 4\sqrt{2\,\mathsf{rank}(\tilde{W})}\|W - \tilde{W}\|_F$.*

*Proof.* The proof closely follows that of Lemma 17 of [Lafond, 2015]. By definition, we have:

$$\Phi_Y^\lambda(\mathbf{X}, \mathbf{W}) - \Phi_Y^\lambda(\mathbf{X}, \tilde{\mathbf{W}}) \leq 0$$

or,

$$\Phi_Y(\mathbf{X}, \mathbf{W}) - \Phi_Y(\mathbf{X}, \tilde{\mathbf{W}}) \leq \lambda(\|\tilde{\mathbf{W}} - \mathbf{W}\|_*)\,.$$

Writing $\mathbf{W} \in \mathbb{R}^{d \times L}$ as $\mathbf{W} = \tilde{\mathbf{W}} + \mathcal{P}_{\tilde{\mathbf{W}}}^\perp(\mathbf{W} - \tilde{\mathbf{W}}) + \mathcal{P}_{\tilde{\mathbf{W}}}(\mathbf{W} - \tilde{\mathbf{W}})$, Lemma 16-(i) of [Lafond, 2015] and triangle inequality together give:

$$\|\mathbf{W}\|_* \geq \|\tilde{\mathbf{W}}\|_* + \|\mathcal{P}_{\tilde{\mathbf{W}}}^\perp(\mathbf{W} - \tilde{\mathbf{W}})\|_* + \|\mathcal{P}_{\tilde{\mathbf{W}}}(\mathbf{W} - \tilde{\mathbf{W}})\|_*,$$

Or,
$$\Phi_Y(\mathsf{X},\tilde{\mathsf{W}}) - \Phi_Y(\mathsf{X},\mathsf{W}) \geq \lambda(\|\mathcal{P}^{\perp}_{\tilde{\mathsf{W}}}(\mathsf{W}-\tilde{\mathsf{W}})\|_* + \|\mathcal{P}_{\tilde{\mathsf{W}}}(\mathsf{W}-\tilde{\mathsf{W}})\|_*) \, . \tag{25}$$

Note that by convexity of $\Phi_Y$:
$$\Phi_Y(\mathsf{X},\tilde{\mathsf{W}}) - \Phi_Y(\mathsf{X},\mathsf{W}) \leq \left\langle \nabla\Phi_Y(\mathsf{X},\tilde{\mathsf{W}}), \mathsf{X}\tilde{\mathsf{W}} - \mathsf{X}\mathsf{W} \right\rangle = \left\langle \mathsf{X}^T\nabla\Phi_Y(\mathsf{X},\tilde{\mathsf{W}}), \tilde{\mathsf{W}} - \mathsf{W} \right\rangle,$$

By trace inequality, we have:
$$\Phi_Y(\mathsf{X},\tilde{\mathsf{W}}) - \Phi_Y(\mathsf{X},\mathsf{W}) \leq \|\mathsf{X}^T\nabla\Phi_Y(\mathsf{X},\tilde{\mathsf{W}})\|_2 \|\tilde{\mathsf{W}} - \mathsf{W}\|_* \leq \frac{\lambda}{2}\|\tilde{\mathsf{W}} - \mathsf{W}\|_*$$

where the last inequality is by assumption, $\|\mathsf{X}^T\nabla\Phi_Y(\mathsf{X},\tilde{\mathsf{W}})\|_2 \leq \lambda/2$. The last term in the above inequality can be bounded by $\frac{\lambda}{2}\left(\|\mathcal{P}^{\perp}_{\tilde{\mathsf{W}}}(\mathsf{W}-\tilde{\mathsf{W}})\|_* + \|\mathcal{P}_{\tilde{\mathsf{W}}}(\mathsf{W}-\tilde{\mathsf{W}})\|_*\right)$. Together with (25), we get the first part of the Lemma. We can now conclude the proof of part two using identical arguments as in Lemma 17 of [Lafond, 2015]. $\square$

**Lemma 6.** *Let $\sigma_{\min}(\mathsf{X})$ denote the smallest singular value of $\mathsf{X}$. Then for any $W, \tilde{W}$, Then:*
$$\|XW - X\tilde{W}\|_F^2 \geq \sigma_{\min}^2(\mathsf{X})\|W - \tilde{W}\|_F^2.$$

*Proof.* Observe that $\|\mathsf{X}(\mathsf{W} - \tilde{\mathsf{W}})\|_F^2 = \mathbf{trace}\big(\mathsf{X}(\mathsf{W} - \tilde{\mathsf{W}})(\mathsf{W} - \tilde{\mathsf{W}})^T\mathsf{X}^T\big) = \mathbf{trace}\big((\mathsf{W} - \tilde{\mathsf{W}})(\mathsf{W} - \tilde{\mathsf{W}})^T\mathsf{X}^T\mathsf{X}\big) \geq \sigma_{\min}(\mathsf{X}^T\mathsf{X})\mathbf{trace}\big((\mathsf{W} - \tilde{\mathsf{W}})(\mathsf{W} - \tilde{\mathsf{W}})^T\big) = \sigma_{\min}(\mathsf{X})^2\|\mathsf{W} - \tilde{\mathsf{W}}\|_F^2.$ $\square$

**Lemma 7.** *Let $X \in \mathbb{R}^{n\times d}$ be a matrix with rows sampled from sub-Gaussian distribution satisfying Assumption 1. Furthermore, choose:*
$$n \geq C'd \, .$$
*Then, with probability at least $1 - 2e^{-d}$, each of the following statements is true:*
$$\sigma_{\max}(X^TX) \leq \bar{C}n,$$
$$\sigma_{\min}(X^TX) \geq \underline{C}n,$$
*where $C', \bar{C}$ and $\underline{C}$ are absolute constants that depend only on the parameters $K$ and $\Sigma$ of the sub-Gaussian distribution.*

*Proof.* Using Lemma 16 of Bhatia et al. [2015b], we have for any $\delta > 0$, with probability at least $1 - \delta$, each of the following statements hold:
$$\sigma_{\max}(\mathsf{X}^T\mathsf{X}) \leq \sigma_{\max}(\Sigma) \, . \, n + C_K\sqrt{dn} + t\sqrt{n},$$
$$\sigma_{\min}(\mathsf{X}^T\mathsf{X}) \geq \sigma_{\min}(\Sigma) \, . \, n - C_K\sqrt{dn} - t\sqrt{n},$$
where $t = \sqrt{\frac{1}{c_K}\log\frac{2}{\delta}}$, and $c_K, C_K$ are absolute constants that depend only on the sub-Gaussian norm $K$ of the distribution $\mathbb{P}_{\mathcal{X}}$. Now, choosing $\delta = 2e^{-d}$ or $\log(2/\delta) = d$, we have:
$$C_K\sqrt{dn} + t\sqrt{n} = C_K\sqrt{dn} + \sqrt{\frac{1}{c_K}dn} = \sqrt{dn}\left(C_K + \sqrt{\frac{1}{c_k}}\right).$$

For ease, define $C'_K := C_K + \sqrt{\frac{1}{c_k}}$. Now, choosing $n \geq \left(\frac{C'_K}{\sigma_{\min}(\Sigma)}\right)^2 \, . \, d$, and substituting above we have:
$$C_K\sqrt{dn} + t\sqrt{n} \leq \frac{1}{2}\sigma_{\min}(\Sigma) \, . n.$$

Therefore:
$$\sigma_{\max}(\mathsf{X}^T\mathsf{X}) \leq \left(\sigma_{\max}(\Sigma) + \frac{1}{2}\sigma_{\min}(\Sigma)\right)n,$$
$$\sigma_{\min}(\mathsf{X}^T\mathsf{X}) \geq \frac{1}{2}\sigma_{\min}(\Sigma) \, . \, n.$$

The proof is complete. $\square$

**Proof of Lemma 1**

Let $H$ denote the matrix with $h_{ij} = y_{ij} - G'(\langle \mathbf{x}_i, \mathbf{w}_j^* \rangle)$. Let $\mathbf{h}^i$ denote the $i$th row of $H$. Let $\mathcal{P}_\Omega(H)$ denote the projection of $H$ onto the observed indices $\Omega$. Let $\Omega_i$ denote the observed indices in row $i$ of $\mathsf{Y}$. For a vector $\mathbf{v}$, let $\mathbf{v}_{\Omega_i}$ denote its projection onto the observed indices $\Omega_i$.

Fix $\mathbf{u} \in \mathbb{R}^d$ and $\mathbf{v} \in \mathbb{R}^L$. Define $a_i = \mathbf{x}_i^T \mathbf{u}$ and $b_i = \langle \mathbf{v}_{\Omega_i}, \mathbf{h}_{\Omega_i}^i \rangle$. We have:

$$\frac{1}{|\Omega|} \mathbf{u}^T \mathsf{X}^T \mathcal{P}_\Omega(H) \mathbf{v} = \frac{1}{|\Omega|} \sum_{i=1}^n a_i b_i$$

$$= \frac{1}{|\Omega|} \sum_{i=1}^n \|\mathbf{v}_{\Omega_i}\|_2 \cdot a_i \frac{b_i}{\|\mathbf{v}_{\Omega_i}\|_2}.$$

Consider $b_i = \sum_{(i,j) \in \Omega} v_j h_{ij}$. Note that $h_{ij}$'s are sub-Gaussian random variables with sub-Gaussian norm $\alpha$. Using Lemma 5.9 of Vershynin [2010], we have $b_i$ is sub-Gaussian with norm $\|\mathbf{v}_{\Omega_i}\|_2 \alpha$. In turn, this implies, $\frac{b_i}{\|\mathbf{v}_{\Omega_i}\|_2}$ is sub-Gaussian with sub-Gaussian norm $\alpha$. Therefore, $\frac{a_i b_i}{\|\mathbf{v}_{\Omega_i}\|_2}$ is $\alpha$-subexponential. Applying Proposition 5.16 of Vershynin [2010], we have, with probability at least $1 - \delta$,

$$\frac{1}{|\Omega|} \sum_{i=1}^n \|\mathbf{v}_{\Omega_i}\|_2 \cdot a_i \frac{b_i}{\|\mathbf{v}_{\Omega_i}\|_2} \leq \frac{c \cdot \alpha}{|\Omega|} \left( \sqrt{\sum_{i=1}^n \|\mathbf{v}_{\Omega_i}\|^2} \sqrt{\log \frac{2}{\delta}} + \max_{i \in [n]} \|\mathbf{v}_{\Omega_i}\|^2 \log \frac{2}{\delta} \right).$$

for some absolute constant $c$. Noting that: $\|\mathbf{v}\|_2 = 1$ and for any $j \in [L]$, $|\{i : (i,j) \in \Omega\}| \leq \frac{c' \cdot |\Omega|}{L}$ for some constant $c'$, we have, with probability at least $1 - \delta$,

$$\frac{1}{|\Omega|} \sum_{i=1}^n \|\mathbf{v}_{\Omega_i}\|_2 \cdot a_i \frac{b_i}{\|\mathbf{v}_{\Omega_i}\|_2} \leq \frac{c \cdot \alpha}{|\Omega|} \left( \sqrt{\frac{c' \cdot |\Omega|}{L}} \sqrt{\log \frac{2}{\delta}} + \log \frac{2}{\delta} \right).$$

We conclude the proof by a covering argument: Taking a union bound over $\epsilon$-ball of $\mathbf{u}$ and $\mathbf{v}$, we have, with probability at least $1 - (d + L)^{-1}$:

$$\left\| \mathsf{X}^T \nabla \Phi_Y(\mathsf{X}, \mathsf{W}^*) \right\|_2 \leq \frac{c \cdot \alpha}{|\Omega|} \left( \sqrt{\frac{c' \cdot |\Omega|}{L}} \sqrt{d + L} + d + L \right).$$

Assuming $d \leq L$ and $|\Omega| \geq (L + d)$, the proof is complete.