[Reviews · NeurIPS 2016]

Reviewer 1

Summary

The paper introduces a new general algorithm for learning when the objective function cannot be decomposed into a sum of losses over individual data points, such as F1 score. The algorithm works in settings such as collaborative filtering and multilabel classification with missing labels. Regret bounds and empirical evaluation show that the algorithm offers improvements for example in the case where the overall label matrix has low rank.

Qualitative Assessment

*** Reply to rebuttal I'm still unconvinced about some technical issues. However I understand that space is limited in the paper and in rebuttal, so I'm willing to give the authors the benefit of doubt. Using the same notation for an empirical quantity and its expectation, in a proof about sample complexity, is really too confusing and needs to be fixed before the proof can be verified. Regarding the assumptions, unless I'm missing something, gamma geq b_0 + min(b_11, b_01,b_10,b_00) gives zero to some interesting measures. *** The paper builds closely upon recent work in the area, but the new algorithm and its analysis are interesting enough. The theoretical discussion is a bit vague about some assumptions and the effect of some parameters of the problem. The experiments compare a variety of data sets but only one competing algorithm. I couldn't really understand the definition of the metric Psi and the related manipulations. On page 3, the quantities TP etc. are defined as true expectations with respect to all relevant randomness. If that's so, then what is for example Y_Omega that appears inside Psi (and thus inside expectation) for example in Lemma 4? Also, Psi is directly optimised in the algorithm, so presumably some empirical estimate should be used instead. However, in proof of Lemma 2, we need that TP_i = TP etc. which is not necessarily true for the obvious empirical estimates. This really needs to be clarified. I would also like to see some comments on the assumptions in particular regarding the constant gamma. Assuming the denominators to be bounded away from zero seems non-trivial, in particular in cases where the bound is assumed to hold for each row or column individually. Another assumption related to Psi is Psi^*c_2-c_1 > 0 etc. (line 387 in proof of Lemma 2). Do these assumptions hold for your example cases? And I don't see how constants c_1 etc. can be obtained that depend on just Psi (and not for example true class distribution). Minor comments: The notation in Section 4.1 is a bit confusing. You generally use y to denote values in {0,1}, but then for example (1-yt)^2 is not good loss functions (for y=0 it's independent of t). line 202: You have already used gamma in another meaning (p. 3). line 215: Some typo in "C'. d"? What do you mean by "numerical constant" in the statements of theorems? I assume it's not the same as "absolute constant," since you use also that term, and looking at the proofs, your "numerical" constants don't seem "absolute." After Theorem 2: What exactly do you mean by "starts converging?" In Lemma 1 and Theorem 4, you should say there is a constant such that the claim hold with high probability, rather than that with high probability a constant exists. Caption of Table 1: Based on just this, "consistently across data sets" seems a bit strong. line 361: "Risk" missing on rhs in definition of Risk*_alpha line 366: I don't think the definition of strongly proper can be found in Reid and Williamson. lines 425 and 443: one ")" missing Theorem 5: What is mu in the bound? line 432: The proof of Theorem 2 is complete only for one particular value of delta. line 438: I don't see how Corollary 2 would follow directly from Theorem 5. proof of Lemma 1: what is c'? [I didn't read Appendix B.]

Confidence in this Review

2-Confident (read it all; understood it all reasonably well)


Reviewer 2

Summary

The paper extends existing results on regret bounds for complex performance measures. The authors consider a learning problem with missing labels, i.e., no all labels are specified in the training set. This is a very up-to-date problem. The algorithmic solution is rather simple and follows from previous results. The theoretical results are also based on previous works, but give more useful bounds by capturing the rank of optimal model and imposing additional assumption on the model.

Qualitative Assessment

The paper is certainly very interesting, well-written, but also very dense. Below I list some minor comments: - The sampling model given in Section 2 could be better motivated. The authors do not discuss any alternative approach and do not exhaustively justify the use of the given model. - Assumption 1 and 2 in Section 4.2.2 should also be better motivated. - The PU learning in case of extreme classification could be discussed in more depth. - In the experiment on synthetic data only a noise-free setting is considered. I suppose that in this case models for different performance measures (like accuracy and the F-measure) will be very similar to each other. I suppose that using more sophisticated models will give more interesting insight into the method.

Confidence in this Review

2-Confident (read it all; understood it all reasonably well)


Reviewer 3

Summary

This paper suggests a theoretically grounded learning algorithm for collaborative filtering and matrix completion, in a general formulation that embraces also multilabel classification and PU learning. Roughly speaking, the algorithm is provided with a label matrix of n rows (the amount of training examples) and L columns (the number of labels). Some matrix entries are missing, and the algorithm must "fill in the blanks" while minimizing a chosen metric. The paper provides both learning regret bound and empirical experiments evaluating their algorithm.

Qualitative Assessment

First of all, I confess that I am not familiar with collaborative filtering and matrix completion literature. That being said, the learning algorithm proposed in the paper is simple and intuitive, supported by a sound theory, and the empirical results are convincing (at least compared to Koyevo et al. 2015 benchmark). From my point of view, it seems like a very nice bridge between theory and practice. I also like that the approach is general enough to embrace many kinds of metrics and apply to different learning frameworks. The paper is globally well written. However, I have the following questions about the proposed algorithm: - Equation (6): How \gamma is defined? - Step 3: Is the maximized expression is concave according to \gamma? - The authors keep saying that they consider "strongly proper loss". I would like the authors to define exactly this term, and explain why it is mandatory. - Do this "strongly proper loss" \ell needs to be related to the choice of the metric \Phi? Even if the regret bounds hold, my intuition that good learning is only achievable with a good (\ell, \Phi) combination. In other words, it may be counterintuitive to minimize a loss when the object of interest is a (perhaps unrelated) metric.

Confidence in this Review

2-Confident (read it all; understood it all reasonably well)


Reviewer 4

Summary

The authors propose a framework for optimizing non-decomposable performance metrics using a consistent plug-in estimator could be applied to the problems of multi-labeling (with missing labels) as well as collaborative filtering. The central idea is based on previous work proving that, for a large class of performance metrics in the context of multi-label classification, thresholding the class probability vector leads to bayes-optimal estimators. The novelty of this work is the extension of this framework to include multi-label classification with missing labels which can be extended to Collaborative Filtering. Consequently, theoretical regret bounds are derived for metrics such as the F-measure and the resulting inference scheme is applied to real-world data sets in the context of multi-label classification.

Qualitative Assessment

The proposed algorithm outperformed a baseline in 3 out of the 5 datasets and was not tested for collaborative filtering which was a large part of the theoretical derivation. The main novelty seems to be a rather straight-forward extension of an existent framework by Koyejo. The main addition is the inclusion of a low-rank estimation of the parameter matrix W. While this is a valid assumption and is being used extensively in the context of Collaborative Filtering as well as dimensionality reduction, it wasn't clear how this is the only change required for this framework to support missing labels. The empirical analysis is also lacking as it only offers an insight on the improvement as compared to the expectedly lacking Koyejo framework (in that it doesn't assume a low-rank structure for the observation matrix). Accordingly, the paper doesn't explain if these regret bounds and the resulting precision metrics offer any improvement to popular Collaborative Filtering algorithms (PMF, ExpoMF, Poisson-NMF, HPF, BPR, etc...). Therefore, the practical motivations seem lacking since the main point of comparison on the CF setting are the works of Lafond, 2015 and Yu, 2014 who are hardly representative of the recent development in the domain. The writing was rather informal with a frequent use of "like, say". For example, the author would write: "Many times, like say in collaborative filtering, features..." This, in addition to missing articles and awkward phrasing, made it harder to follow the line of thought of this work.

Confidence in this Review

3-Expert (read the paper in detail, know the area, quite certain of my opinion)


Reviewer 5

Summary

This paper deals with the practically important setting when the performance metrics (like the F1 measure) is non-decomposable over labels and the training data has missing labels. The regret or generalization error in the given metric is bounded ultimately by estimation error of certain underlying parameters. Regret bounds is derived under three settings: a) collaborative filtering, b) multilabel classification, and c) PU (positive-unlabeled) learning.

Qualitative Assessment

It is an interesting machine learning theory paper. The author claims that for each of the above problems, they can obtain precise non-asymptotic regret bound which is small even when a large fraction of labels is missing. It could be better and reasonable to compare the derived bound with other related works.

Confidence in this Review

2-Confident (read it all; understood it all reasonably well)


Reviewer 6

Summary

The paper proposes a method to optimize non-decomposable loss such as F-measure with missing entries. It adds a trace norm regularizer to deal with the missing entries. A threshold is then learned for all the labels to differentiate relevant labels from irrelevant ones. The paper theoretically bounds the regret by \lambda-strongly proper loss and further bounds the \lambda-strongly proper loss for three different applications. Experimental results verify the effectiveness of the proposed method.

Qualitative Assessment

The paper solves the missing labels problem for non-decomposable loss such as F-measure. Regret bounds for three applications are proposed. Experiments verify the merit of the proposal on both accuracy and F1. The paper is not quite clear in the following aspects 1) Compared to classical matrix completion algorithms solving the trace norm regularization, the proposed algorithm needs to bound \|XW\|_\infty. Can detailed optimization procedure be given on how to bound the infinity norm of XW? 2) Theoretically the \lambda-strongly proper loss should be 1-lipschitz. Does the definition of \lambda-strongly proper loss imply it is 1-lipschitz? 3) How to get the best threshold \theta? By brute force searching of all possible ones using validation set? Will it be low-efficiency?

Confidence in this Review

2-Confident (read it all; understood it all reasonably well)